# PROTORES: PROTO-RESIDUAL NETWORK FOR POSE AUTHORING VIA LEARNED INVERSE KINEMATICS

Boris N. Oreshkin*    Florent Bocquelet    Félix G. Harvey    Bay Raitt    Dominic Laflamme

## ABSTRACT

Our work focuses on the development of a learnable neural representation of human pose for advanced AI assisted animation tooling. Specifically, we tackle the problem of constructing a full static human pose based on sparse and variable user inputs (*e.g.* locations and/or orientations of a subset of body joints). To solve this problem, we propose a novel neural architecture that combines residual connections with prototype encoding of a partially specified pose to create a new complete pose from the learned latent space. We show that our architecture outperforms a baseline based on Transformer, both in terms of accuracy and computational efficiency. Additionally, we develop a user interface to integrate our neural model in Unity, a real-time 3D development platform. Furthermore, we introduce two new datasets representing the static human pose modeling problem, based on high-quality human motion capture data, which will be released publicly along with model code.

## 1 INTRODUCTION

Modeling human pose and learning pose representations have received increasing attention recently due to their prominence in applications, such as computer graphics and animation (Harvey et al., 2020; Xu et al., 2020); immersive augmented reality (Facebook Reality Labs, 2021; Capece et al., 2018; Lin, 2019; Yang et al., 2021); entertainment (McDonald, 2018; Xpire, 2019); sports and wellness (Rosenhahn et al., 2008; Kim et al., 2021) as well as human machine interaction (Heindl et al., 2019; Casillas-Perez et al., 2016; Schwarz et al., 2014) and autonomous driving (Kumar et al., 2021). In the gaming industry, state-of-the-art real-time pose manipulation tools, such as CCD (Kenwright, 2012), FABRIK (Aristidou & Lasenby, 2011) or FinalIK (RootMotion, 2020), are popular for rapid execution and rely on forward and inverse kinematics models defined via non-learnable kinematic equations. *Inverse kinematics* (IK) is the process of computing the internal geometric parameters of a kinematic system resulting in the desired configuration (e.g. global positions) of system's joints (Paul, 1992). *Forward kinematics* (FK) refers to the use of the kinematic equations to compute the positions of joints from specified values of internal geometric parameters. While being mathematically accurate, non-learnable IK models do not attempt reconstructing plausible human poses from underconstrained solutions derived from sparse constraints (e.g. positions of a small subset of joints).

In this paper we develop a neural modeling approach to reconstruct full human pose from a sparse set of constraints supplied by a user, in the context of pose authoring and game development. We bridge the gap between skeleton-aware human pose representation based on IK/FK ideas and the neural embedding of human pose. Our approach effectively implements a learnable model for skeleton IK, mapping desired joint configuration into predictions of skeleton internal parameters (local rotations), learning the statistics of natural poses using datasets derived from high-quality motion capture (MOCAP) sequences. The approach, which we call ProtoRes, models the semantics of joints and their interactions using a novel prototypical residual neural network architecture. Inspired by prototypical networks, which showed that one semantic class can be represented by the prototype (mean) of a few examples (Snell et al., 2017), we extend it using a multi-block residual approach: the final pose embedding is a mean across embeddings of sparse constraints and across partial pose predictions produced in each block. We show that in terms of the pose reconstruction accuracy, ProtoRes outperforms existing gaming industry tools such as FinalIK, as well as out-of-the-box machine-learning solution based on Transformer (Vaswani et al., 2017), which also happens to be 10 times less effective in terms of training speed than the proposed architecture.

---

*All authors are with Unity Labs; correspondence to: B.N. Oreshkin, `boris.oreshkin@gmail.com`

Finally, we develop user-facing tools that integrate learned ProtoRes pose representation in the Unity game engine (see Fig. 1). This provides convincing qualitative examples of solutions to the problem of the AI assisted human pose authoring. At the qualitative level, getting traditional workflows to behave the way ProtoRes does would require one to use many techniques in tandem, including IK, FK, layered animation pose libraries, along with procedural rigs encoding explicit heuristics. The process would be highly labor-intensive even for an experienced user while the results would still be of variable fidelity depending on their skill. This is because traditional rigs have no bias towards realistic poses and only allow exploring a limited linear latent space defined by uniform interpolation of a heuristic constraint system. ProtoRes forms a foundation that allows any junior or indie/studio user to bypass these existing complexities and create entirely new workflows for meaningfully exploring learned latent space using a familiar yet far more powerful way. We believe that our model and tools will help speed up the animation process, simplify and democratize game development.

## 1.1 BACKGROUND

We consider the full-body pose author-ing animation task depicted in Fig. 1. The animator provides a few inputs, which we call *effectors*, that the tar-get pose has to respect. For example, the look-at effector specifies that the head should be facing a specific di-rection, the positional effectors pin some joints to specific locations in global space and rotational effectors constrain the world-space rotations of certain joints. We assume that the an-imator can generate arbitrary number of such effectors placed on any skele-tal joint (one joint can be driven by more than one effector). The task of the model is to combine all the infor-mation provided via effectors and gen-

Figure 1: ProtoRes completes full human pose using an arbitrary combination of 3D coordinates, look-at targets and world-space rotations specified by a user. **The animation is best viewed in Adobe Reader.**

erate a plausible full-body pose respecting provided effector constraints. We define the full-body pose as the set of all kinematic parameters necessary to recreate the appearance of the body in 3D.

We assume that each effector can be represented in the space $\mathbb{R}^{d_{\mathrm{eff}}}$, where $d_{\mathrm{eff}}$ is taken to be maximum over all effector types. Suppose we have 3D position and 6D rotation effectors: $d_{\mathrm{eff}}$ is 6. In position effectors, 3 extra values are 0. We formulate the pose authoring problem as learning the mapping $\Upsilon_{\boldsymbol{\theta}} : \mathbb{R}^{N \times d_{\mathrm{eff}}} \to \mathbb{R}^{d_{\mathrm{kin}}}$ with learnable parameters $\boldsymbol{\theta} \in \Theta$. $\Upsilon_{\boldsymbol{\theta}}$ maps the input space $\mathbb{R}^{N \times d_{\mathrm{eff}}}$ of variable dimensionality $N$ (the number of effectors is not known apriori) to the space $\mathbb{R}^{d_{\mathrm{kin}}}$, containing all kinematic parameters to reconstruct full-body pose. A body with $J$ joints can be fully defined using a tree model with 6D local rotation per joint and 3D coordinate for the root joint, in which case $d_{\mathrm{kin}} = 6J + 3$, assuming fixed bone lengths. Given a dataset $\mathcal{D} = \{\mathbf{x}_i, \mathbf{y}_i\}_{i=1}^{M}$ of poses containing pairs of inputs $\mathbf{x}_i \in \mathbb{R}^{N \times d_{\mathrm{eff}}}$ and outputs $\mathbf{y}_i \in \mathbb{R}^{d_{\mathrm{kin}}}$, $\Upsilon_{\boldsymbol{\theta}}$ can be learned by minimizing empirical risk:

$$\Upsilon_{\boldsymbol{\theta}} = \arg\min_{\boldsymbol{\theta} \in \Theta} \frac{1}{M} \sum_{\mathbf{x}_i, \mathbf{y}_i \in \mathcal{D}} L(\Upsilon_{\boldsymbol{\theta}}(\mathbf{x}_i), \mathbf{y}_i) \tag{1}$$

## 1.2 RELATED WORK

**Joint representations.** Representing pose via 3D joint coordinates (Cheng et al., 2021; Cai et al., 2019; Khapugin & Grishanin, 2019) is sub-optimal as it does not enforce fixed-length bones, nor specifies joint rotations. Predicting joint rotations automatically satisfies bone length constraints and adequately models rotations (Pavllo et al., 2018), which is crucial in downstream applications, such as deforming a 3D mesh on top of the skeleton, to avoid unrealistic twisting. This is viable via skeleton representations based on Euler angles (Han et al., 2017), rotation matrices (Zhang et al., 2018) and quaternions (Pavllo et al., 2018). In this work, we use the two-row 6D rotation matrix representation that addresses the continuity issues reminiscent of the other representations (Zhou et al., 2019).

**Pose modeling architectures.** Multi-Layer Perceptrons (MLPs) (Cho & Chen, 2014; Khapugin & Grishanin, 2019; Mirzaei et al., 2020) and kernel methods (Grochow et al., 2004; Holden et al., 2015) have been used to learn single pose representations. Beyond single pose, skeleton moving through time can be modeled as a spatio-temporal graph (Jain et al., 2016) or as a graph convolution (Yan et al., 2018; Mirzaei et al., 2020). A common limitation of these approaches is their reliance on a fixed set of inputs, whereas our architecture is specifically designed to handle sparse variable inputs.

**Pose prediction from sparse constraints.** Real-time methods based on nearest-neighbor search, local dynamics and motion matching have been used on sparse marker position and accelerometer data (Tautges et al., 2011; Riaz et al., 2015; Chai & Hodgins, 2005; Büttner & Clavet, 2015). MLPs and RNNs have been used for real-time processing of sparse signals such as accelerometers (Huang et al., 2018; Holden et al., 2017; Starke et al., 2020; Lee et al., 2018) and VR constraints (Lin, 2019; Yang et al., 2021). These approaches rely on the past pose information to disambiguate next frame prediction and as such are not applicable to our problem, in which only current pose constraints are available. Iterative IK algorithms such as FinalIK (RootMotion, 2020) have been popular in real-time applications. FinalIK works by setting up multiple IK chains for each limb of the body of a predefined human skeleton and for a fixed set of effectors. Several iterations are executed to solve each of these chains using a conventional bone chain IK method, e.g. CCD. In FinalIK, the end effector (hands and feet) can be positioned and rotated, while mid-effectors (shoulders and thighs) can only be positioned. Effectors can have a widespread effect on the body via a hand-crafted pulling mechanism that gives a different weight to each chain. This and similar tools suffer from limited realism when used for human full-body IK, as they are not data-driven. Learning-based methods strive to alleviate this by providing learned model of human pose. Grochow et al. (2004) proposed a kernel based method for learning a pose latent space in order to produce the most likely pose satisfying sparse effector constraints via online constrained optimization. The more recent commercial tool Cascadeur uses a cascade of several MLPs (each dealing with fixed set of positional effectors: 6, 16, 28) to progressively produce all joint positions without respecting bone constraints (Khapugin & Grishanin, 2019; Cascadeur, 2019). Unlike our approach, Cascadeur cannot handle arbitrary effector combinations, rotation or look-at constraints and requires post processing to respect bone constraints.

**Permutation invariant architectures.** Models for encoding unstructured variable inputs have been proposed in various contexts. Attention models (Bahdanau et al., 2015) and Transformer (Vaswani et al., 2017) have been proposed in the context of natural language processing. Prototypical networks (Snell et al., 2017) used average pooled embedding to encode semantic classes via a few support images in the context of few-shot image classification. Maxpool representations over variable input dimension were proposed by Qi et al. (2017) as PointNet and Zaheer et al. (2017) as DeepSets for segmentation and classification of 3D point clouds, image tagging, set anomaly detection and text concept retrieval. Niemeyer et al. (2019) further generalized the PointNet by chaining the basic maxpool/concat PointNet blocks resulting in ResPointNet architecture.

## 1.3 SUMMARY OF CONTRIBUTIONS

The contributions of our paper can be summarized as follows.

- We define the 3D character posing task and publicly release two associated benchmarks.

- We show that a learned inverse kinematics solution can construct better poses, qualitatively and quantitatively, compared to a non-learned approach.

- We extend existing architectures with (i) semantic conditioning of joint ID and type at the input, (ii) novel residual scheme involving prototype subtraction and accumulation across blocks, as opposed to maxpool/concat daisy chain of ResPointNet, (iii) two-stage architecture with computationally efficient residual decoder that improves accuracy at smaller computational cost, as opposed to the naive final linear projection approach of PointNet and ResPointNet, and (iv) two-stage decoder design.

- We propose a novel look-at loss function and a novel randomized weighting scheme combining randomly generated effector tolerance levels and effector noise to increase the effectiveness of multi-task training.

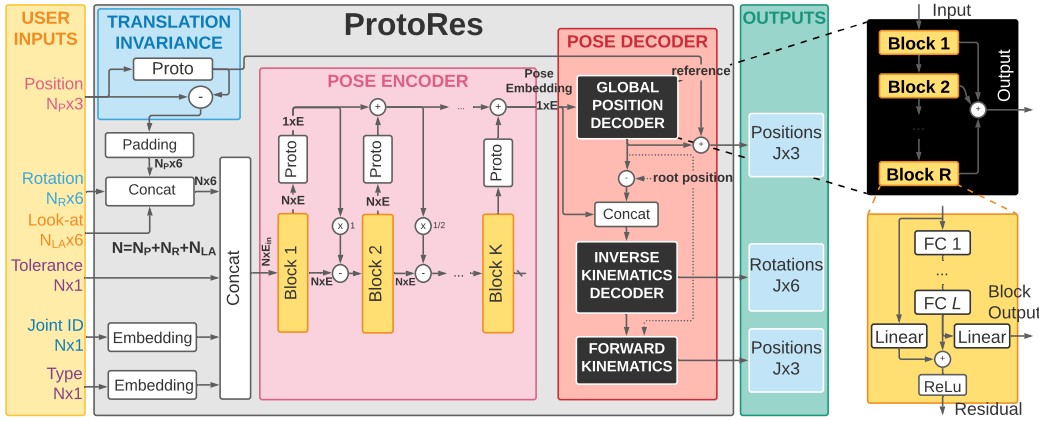

Figure 2: ProtoRes follows the encoder-decoder pattern and produces predictions in three steps. First, the variable number and type of user supplied inputs are processed for translation invariance and embedded. Second, proto-residual encoder transforms the pose specified via effectors into a pose embedding. Finally, the pose decoder expands the pose embedding into the full-body pose representation including local rotation and global position of each joint.

## 2  PROTORES

ProtoRes, shown in Fig. 2, follows the encoder-decoder pattern, unlike PointNet and ResPointNet that use a linear layer as a decoding mechanism (Qi et al., 2017; Niemeyer et al., 2019). The encoder has to deal with $N$ effectors, whereas the decoder processes the collapsed representation of the pose and is therefore $N$ times more compute efficient. Adding more decoder blocks thus results in accuracy gains at a fraction of compute cost. Below, we describe the rest of architecture in more detail.

**User inputs.** ProtoRes accepts position (3D coordinates), rotation (6D representation of Zhou et al. (2019)) and look-at (3D target position and 3D local facing direction) effectors. All positions are re-referenced relative to the centroid of the positional effectors to achieve translation invariance. Translation invariance simplifies the handling of poses in global space removing the need in universal reference frame, which is tricky to define. Each effector is further characterized by a positive tolerance value. Smaller tolerance implies that the effector has to be more strictly respected in the reconstructed pose. Moreover, the input includes effector semantics encoded via joint ID, an integer in $[0, J]$, and effector type, indicating a positional (0), rotational (1) or look-at (2) effector. Unlike e.g. PointNet (Qi et al., 2017) acting on dense extensive point clouds, ProtoRes acts on sparse inputs that barely provide enough information about the full pose. Therefore, it is critical to provide the semantic information on whether the given effector affects hands or feet and whether this is a positional or a rotational effector, for example. Type and joint ID variables are embedded into continuous vectors and concatenated with effector data, resulting in the encoder input, a matrix $\mathbf{x}_{in} \in \mathbb{R}^{N \times E_{in}}$ with $E_{in}$ corresponding to the combined dimension of all embeddings plus 6D effector data and a scalar for tolerance.

**Pose Encoder** is a two-loop residual network. The first residual loop is implemented inside each block depicted in Fig. 2 (bottom right). The second residual loop shown in Fig. 2 (left) implements the proposed Prototype-Subtract-Accumulate (PSA) residual stacking principle, which we empirically found to outperform ResPointNet's Maxpool-Concat daisy chain proposed by Niemeyer et al. (2019). Next, we first lay out the encoder equations and then describe the motivation behind them in detail. We assume the encoder input to be $\mathbf{x}_1 = \mathbf{x}_{in} \in \mathbb{R}^{N \times E_{in}}$, omitting the batch dimension for brevity, in which case the fully-connected layer $\text{FC}_{r,\ell}$, with $\ell = 1...L$, in the residual block $r$, $r = 1 \ldots R$, with weights $\mathbf{W}_{r,\ell}$ and biases $\mathbf{a}_{r,\ell}$ can be conveniently described as $\text{FC}_{r,\ell}(\mathbf{h}_{r,\ell-1}) \equiv \text{RELU}(\mathbf{W}_{r,\ell}\mathbf{h}_{r,\ell-1} + \mathbf{a}_{r,\ell})$. The prototype layer is defined as $\text{PROTOTYPE}(\mathbf{x}) \equiv \frac{1}{N}\sum_{i=1}^{N}\mathbf{x}[i,:]$. The pose encoder is then described as:

$$\mathbf{x}_r = \text{RELU}[\mathbf{b}_{r-1} - 1/(r-1) \cdot \mathbf{p}_{r-1}], \tag{2}$$

$$\mathbf{h}_{r,1} = \text{FC}_{r,1}[\mathbf{x}_r], \ \ldots, \ \mathbf{h}_{r,L} = \text{FC}_{r,L}[\mathbf{h}_{r,L-1}], \tag{3}$$

$$\mathbf{b}_r = \text{RELU}[\mathbf{L}_r\mathbf{x}_r + \mathbf{h}_{r,L}], \ \mathbf{f}_r = \mathbf{F}_r\mathbf{h}_{r,L}, \tag{4}$$

$$\mathbf{p}_r = \mathbf{p}_{r-1} + \text{PROTOTYPE}[\mathbf{f}_r]. \tag{5}$$

Equations (3) and (4) implement the MLP and the first residual loop. The proposed PSA residual mechanism, described in equations (2) and (5), is motivated by the following. First, it implements the inductive bias that the information in individual effectors is only valuable when it is different from what is already stored in the embedding of entire pose. Equation (2) implements this logic by forcing delta-mode in effectors w.r.t. to the pose embedding, $\mathbf{p}_{r-1}$, from the previous block, which additionally creates another residual loop that should facilitate gradient flow. Second, equation (5) collapses the forward encoding of individual effectors into the representation of the entire pose via prototype, which is known to be very effective at representing information from sparse examples (see e.g. Snell et al. (2017)). Finally, the representation of pose is accumulated across residual blocks in (5), effectively implementing skip connections from very early layers. Distant skip connections implemented via concatenation were shown to be effective in DenseNet (Huang et al., 2017). Concatenation based skipping requires additional computation and is most efficient with convolutional networks, in which kernel size can be traded for feature width while implementing distant skip connections, whereas accumulation is more compute efficient in our context.

**Pose Decoder** has two blocks: global position decoder (GPD) and inverse kinematics decoder (IKD). Both rely on the fully-connected residual (FCR) architecture depicted in Fig. 2 (right). GPD unrolls the pose embedding generated by encoder into the unconstrained predictions of 3D joint positions. IKD generates the internal geometric parameters (joint rotations) of the skeleton that are guaranteed to generate feasible joint positions after forward kinematics pass.

**GPD** accepts the encoded pose embedding, $\widetilde{\mathbf{b}}_0 = \mathbf{p}_R \in \mathbb{R}^E$, and produces 3D position predictions $\widetilde{\mathbf{f}}_R \in \mathbb{R}^{3J}$ of all skeletal joints using the FCR whose $r$-th block is described as follows:

$$\mathbf{h}_{r,1} = \text{FC}_{r,1}^{gpd}[\widetilde{\mathbf{b}}_{r-1}], \ldots, \mathbf{h}_{r,L} = \text{FC}_{r,L}^{gpd}[\mathbf{h}_{r,L-1}],$$
$$\widetilde{\mathbf{b}}_r = \text{RELU}[\mathbf{L}_r^{gpd}\widetilde{\mathbf{b}}_{r-1} + \mathbf{h}_{r,L}], \ \widetilde{\mathbf{f}}_r = \widetilde{\mathbf{f}}_{r-1} + \mathbf{F}_r^{gpd}\mathbf{h}_{r,L}. \tag{6}$$

Since GPD produces predictions with no regard to skeleton constraints, its predictions do not respect bone lengths. For the IKD to provide correct rotations, the origin of the kinematic chain in world space must be given, and GPD conveniently provides the prediction of the reference (root) joint.

**IKD** accepts input $\widehat{\mathbf{b}}_0 \in \mathbb{R}^{E+3J}$, consisting of the concatenation of the encoder-generated pose embedding, $\mathbf{p}_R \in \mathbb{R}^E$, and the output of GPD, $\widetilde{\mathbf{f}}_R \in \mathbb{R}^{3J}$. Effectively, the draft pose generated by GPD, is used to condition IKD. We show in Section 3 that this additional conditioning improves accuracy. IKD predicts the 6D angle for each joint, $\widehat{\mathbf{f}}_R \in \mathbb{R}^{6J}$, and its $r$-th block operates as follows:

$$\mathbf{h}_{r,1} = \text{FC}_{r,1}^{ikd}[\widehat{\mathbf{b}}_{r-1}], \ldots, \mathbf{h}_{r,L} = \text{FC}_{r,L}^{ikd}[\mathbf{h}_{r,L-1}],$$
$$\widehat{\mathbf{b}}_r = \text{RELU}[\mathbf{L}_r^{ikd}\widehat{\mathbf{b}}_{r-1} + \mathbf{h}_{r,L}], \ \widehat{\mathbf{f}}_r = \widehat{\mathbf{f}}_{r-1} + \mathbf{F}_r^{ikd}\mathbf{h}_{r,L}. \tag{7}$$

**Forward Kinematics** (FK) pass, described in detail in Appendix A, applies skeleton kinematic equations to the local joint rotations and global root position produced by IKD. For each joint $j$, it produces the global transform matrix $\widehat{\mathbf{G}}_j$ containing the global rotation matrix, $\widehat{\mathbf{G}}_j^{13} \equiv \widehat{\mathbf{G}}_j[1:3, 1:3]$, and the 3D global position, $\widehat{\mathbf{g}}_j = \widehat{\mathbf{G}}_j[1:3, 4]$, of the joint.

## 2.1 LOSSES

We use three losses to train the architecture in a multi-task fashion. The total loss combines loss terms additively with weights chosen to approximately equalize their magnitude orders.

**L2 loss** penalizes the mean squared error between the prediction $\widehat{\mathbf{y}}$ and the ground truth $\mathbf{y}$:

$$\text{MSE}(\mathbf{y}, \widehat{\mathbf{y}}) = \|\mathbf{y} - \widehat{\mathbf{y}}\|_2^2. \tag{8}$$

L2 loss is used to supervise the GPD as well as the IKD output after FK pass. In the latter case it drives IKD to learn to predict rotations that lead to small position errors after FK.

**Geodesic loss** penalizes the errors of the IKD's rotational outputs. It represents the smallest arc (in radians) to go from one rotation to another over the surface of a sphere. The geodesic loss is defined for the ground truth rotation matrix $\mathbf{R}$ and its prediction $\widehat{\mathbf{R}}$ as (see e.g. Salehi et al. (2018)):

$$\text{GEO}(\mathbf{R}, \widehat{\mathbf{R}}) = \arccos\left[(\text{tr}(\widehat{\mathbf{R}}^T\mathbf{R}) - 1)/2\right]. \tag{9}$$

We believe that using a combination of positional and rotational losses is necessary to learn a high-quality pose representation. This is especially important when the task is to reconstruct a sparsely specified pose, giving rise to multiple plausible reconstructions. We argue that a model trained to reconstruct both plausible joint positions and rotations is better equipped to solve the task accurately. Empirical evidence presented in Section 3.4 supports this intuition: a model trained on both L2 and Geodesic generalizes better on both losses than models trained only on one of those terms.

**Look-at loss**, proposed in this paper, enables the "look-at" feature, *i.e.* the ability to orient a joint to face a particular global position (e.g. having the head looking at a given object). It allows the model to align any direction vector $\mathbf{d}_j \in \mathbb{R}^3$ of a joint, expressed in its local frame of reference, towards a global target location $\mathbf{t}$. Given the predicted global transform matrix $\widehat{\mathbf{G}}_j$, look-at loss is defined as:

$$\text{LAT}(\mathbf{t}, \mathbf{d}_j, \widehat{\mathbf{G}}_j) = \arccos\left[\overrightarrow{(\mathbf{t} - \widehat{\mathbf{g}}_j)} \cdot \widehat{\mathbf{G}}_j^{13} \mathbf{d}_j\right]. \tag{10}$$

$\overrightarrow{(\mathbf{t} - \widehat{\mathbf{g}}_j)}$ is a unit-length vector pointing at the target object in world space. $\widehat{\mathbf{G}}_j^{13}$, when multiplied by $\mathbf{d}_j$, represents the global predicted look-at direction. The look-at loss trains the IKD to produce $\widehat{\mathbf{G}}_j^{13}$ consistent with the look-at direction defined by $\mathbf{t}$ and $\mathbf{d}_j$, both provided as network inputs.

## 2.2 TRAINING METHODOLOGY

The training methodology involves techniques to (i) regularize model via rotation and mirror augmentations, (ii) learn handling of sparse inputs and (iii) effectively combine multi-task loss terms.

**Sparse inputs** modeling relies on effector sampling. First, the total number of effectors is sampled uniformly at random in the range [3, 16]. Given the total number of effectors, the effector IDs (one of 64 joints) and types (position, rotation, or look-at) are sampled from the Multinomial without replacement. This induces exponentially large number of effector type and joint permutations, resulting in strong regularizing effects and teaching the network to deal with variable inputs.

**Effector tolerance and randomized loss weighting.** The motivation behind randomized loss weighting is two-fold. First, we empirically find that when a random weight is multiplicatively applied to the respective loss term and its reciprocal is used as one of the network inputs for the corresponding effector, the network learns to respect the tolerance level. When exposed as a user interface feature, it lets the user control the degree of responsiveness of the model to different effectors. We also discovered that this only works when noise is added to the effector value and the standard deviation of the noise is appropriately modulated by tolerance. For example, the noise teaches the model to disregard the effector completely if the tolerance input value corresponds to the high noise variance regime. Second, we notice that the randomized weighting improves multi-task training and generalization performance. In particular, we observe significant competition between rotation and position losses on our task. The introduction of the randomized loss weighting seems to turn the competition into cooperation as our empirical results suggest in Section 3. We implement the randomized loss weighting scheme via the following steps. For each sampled effector, we uniformly sample $\Lambda \in [0, 1]$ treated as the effector tolerance. Given an effector tolerance $\Lambda$, noise with the maximum standard deviation $\sigma_M$ modulated by $\Lambda$ (noise models used for different effector types are described in detail in Appendix B) is added to effector data before feeding them to the neural network:

$$\sigma(\Lambda) = \sigma_M \Lambda^\eta, \tag{11}$$

We use $\eta > 10$ to shape the distribution of $\sigma$ to smaller values. Furthermore, each effector is attached with a randomized loss weight reciprocal to $\sigma(\Lambda)$, capped at $W_M$ if $\sigma(\Lambda) < 1/W_M$:

$$W(\Lambda) = \min(W_M, 1/\sigma(\Lambda)). \tag{12}$$

$\Lambda$ drives network input and $W(\Lambda)$ weighs the loss term affected by the effector.

The detailed procedure to compute the ProtoRes loss based on one batch item is presented in Algorithm 1 of Appendix C and the summary is provided below. First, we sample (i) the number of effectors and (ii) their associated types and IDs. For each effector, we randomly sample the tolerance level and compute the associated noise std and loss weight. Given noise std, an appropriate noise model is applied to generate input data based on effector type as described in Appendix B. Then ProtoRes predicts draft joint positions $\widetilde{\mathbf{f}}_{R,j}$ and local joint rotations $\widehat{\mathbf{R}}_j$. World-space rotations and positions $\widehat{\mathbf{G}}_j$ for all joints $j \in [0, J)$ are computed using FK. We conclude by calculating the individual deterministic and randomized loss terms, whose weighted sum is used for backpropagation.

Table 1: Key quantitative results: ProtoRes vs. baselines. Lower values are better. Inference speed is measured per pose solve on NVIDIA Geforce RTX 2080 Super, Intel Core i7 2.30 Ghz.

| | miniMixamo | | | miniUnity | | | speed, | params |
|---|---|---|---|---|---|---|---|---|
| | $\mathcal{L}_{gpd-L2}^{det}$ | $\mathcal{L}_{ikd-L2}^{det}$ | $\mathcal{L}_{loc-geo}^{det}$ | $\mathcal{L}_{gpd-L2}^{det}$ | $\mathcal{L}_{ikd-L2}^{det}$ | $\mathcal{L}_{loc-geo}^{det}$ | ms | |
| 5-point benchmark | | | | | | | | |
| FinalIK | 5.53e-3 | 8.54e-3 | 0.5287 | 3.76e-3 | 7.83e-3 | 0.5164 | 0.3 | n/a |
| Masked-FCR | 1.30e-3 | 2.49e-3 | 0.2607 | 1.11e-3 | 2.38e-3 | 0.2124 | 5.5 | 43M |
| Transformer | 1.10e-3 | 2.06e-3 | 0.2698 | 0.92e-3 | 1.79e-3 | 0.2138 | 10.6 | 24M |
| ProtoRes | **1.00e-3** | **2.02e-3** | **0.2534** | **0.76e-3** | **1.74e-3** | **0.2037** | 5.5 | 41M |
| Random benchmark | | | | | | | | |
| Masked-FCR | 15.02e-3 | 35.21e-3 | 0.3136 | 1.57e-2 | 3.23e-2 | 0.2694 | - | - |
| Transformer | 1.63e-3 | 4.32e-3 | 0.2599 | 1.27e-3 | 3.49e-3 | 0.2006 | - | - |
| ProtoRes | **1.36e-3** | **4.16e-3** | **0.2381** | **0.93e-3** | **3.28e-3** | **0.1817** | - | - |

## 3 EMPIRICAL RESULTS

Our results demonstrate that (i) ProtoRes reconstructs sparsely defined pose more accurately than existing non-ML IK solution and two ML baselines, (ii) our Prototype-Subtract-Accumulate residual scheme is more effective than the Maxpool-Concat daisy chain of Niemeyer et al. (2019), (iii) two-stage GPD+IKD decoding is more effective than IKD-only decoding, (iv) the proposed randomized loss weighting improves multi-task training, (v) joint Geodesic/L2 loss training is synergetic.

### 3.1 DATASETS

**miniMixamo** We use the following procedure to create our first dataset from the publicly available MOCAP data available from mixamo.com, generously provided by Adobe Inc. (2020). We download a total of 1598 clips and retarget them on our custom 64-joint skeleton using the Mixamo online tool. This skeleton definition is used in Unity to extract the global positions as well as global and local rotations of each joint at the rate of 60 frames per second (total 356,545 frames). The resulting dataset is partitioned at the clip level into train/validation/test splits (with proportion 0.8/0.1/0.1, respectively) by sampling clip IDs uniformly at random. Splitting by clip makes the evaluation framework more realistic and less prone to overfitting: frames belonging to the same clip are often similar. At last, the final splits retain only 10% of randomly sampled frames (miniMixamo has 33,676 frames total after subsampling) and all the clip identification information (clip ID, meta-data/description, character information, etc.) is discarded. This anonymization guarantees that the original sequences from mixamo.com cannot be reconstructed from our dataset, allowing us to release the dataset for reproducibility purposes without violating the original dataset license (Adobe Inc., 2020).

**miniUnity** To collect our second dataset we predefine a wide range of human motion scenarios and hire a qualified MOCAP studio to record 1776 clips (967,258 total frames @60 fps). Data collection details appear in Appendix D. Then we create a dataset of a total of 96,666 subsampled frames following exactly the same methodology that was employed for miniMixamo.

### 3.2 TRAINING AND EVALUATION SETUP

We use Algorithm 1 of Appendix C to sample batches of size 2048 from the training subset, hyperparameters are adjusted on the validation set. The number of effectors is sampled once per batch and is fixed for all batch items to maximize data throughput. We report metrics $\mathcal{L}_{gpd-L2}^{det}$, $\mathcal{L}_{ikd-L2}^{det}$, $\mathcal{L}_{loc-geo}^{det}$, defined in Appendix E and calculated on the test set, using models trained on the training set. $\mathcal{L}_{gpd-L2}^{det}$ is computed only on the root joint. These metrics characterise both the 3D position accuracy and the bone rotation accuracy. The evaluation framework tests model performance on a pre-generated set of seven files containing $6, 7 \ldots 12$ effectors respectively. Metrics are averaged over all files, assessing the overall quality of pose reconstruction in scenario with sparse variable inputs. Tables present results averaged over 4 random seed retries and metrics computed every 10 epochs over last 1000 epochs. Additional details and hyperparameter settings appear in Appendix E.

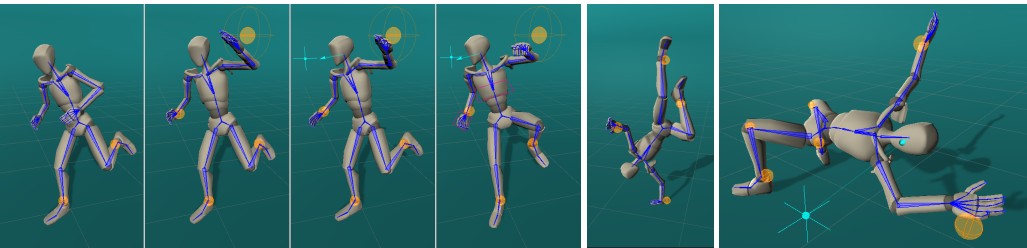

Figure 3: Qualitative posing results. Left 4 poses: adding position, look-at and rotation effectors to specify pose. Right 2 poses: achieving interesting poses with sparse constraints (4 and 7 effectors).

## 3.3 KEY RESULTS

To demonstrate the advantage of the proposed architecture, we perform two evaluations. First, we compare ProtoRes against two ML baselines in the random effector evaluation setup described in Section 3.2. The first baseline, Masked-FCR, is a brute-force unstructured baseline that uses a very wide $J \cdot 3 \cdot 7$ input layer ($J$ joints, 3 effector types, 6D effector data and 1D tolerance) handling all effector permutations. Missing effectors are masked with $3 \cdot J$ learnable 7D placeholders. Masked-FCR has 3 encoder and 6 decoder blocks to match ProtoRes. The second baseline is based on the Transformer encoder (Vaswani et al. (2017), see Appendix G for architecture and hyperparameter settings). The bottom of Table 1, summarizing this study, shows clear advantage of ProtoRes w.r.t. both baselines. Additionally, training Transformer on NVIDIA M40 24GB GPU for 40k epochs of miniMixamo takes 1055 hours (batch size 1024 to fit Transformer in GPU memory), whereas training ProtoRes takes 106 hours. ProtoRes is clearly more compute efficient.

Second, Table 1 (top) compares ProtoRes against a non-ML IK solution FinalIK (RootMotion, 2020), as well as Transformer and Masked-FCR, on a 5-point evaluation benchmark. The 5-point benchmark tests the reconstruction of the full pose from five position effectors: chest, left and right hands, left and right feet. It is chosen, because generating the exponentially large number of FinalIK configurations to process all heterogeneous effector combinations in the random benchmark is not feasible. Note that the 5-point benchmark and random benchmark results are not directly comparable. We can see that all ML methods significantly outperform FinalIK in reconstruction accuracy, ProtoRes being the best overall (please also refer to qualitative analysis results in Appendices J, L.4, L.8). Clearly, ML methods learn the right inductive biases from the data to solve the ill-posed sparse input pose reconstruction problem, unlike the pure non-learnable IK method FinalIK.

Third, qualitative posing results are shown in Fig. 3, demonstrating that visually appealing poses can be obtained with small number of effectors (4 and 7 effectors for the two poses on the right). The left 4 poses demonstrate how pose can be refined successively by adding more effectors. Please refer to Appendix L and supplementary videos for more demonstration examples.

## 3.4 ABLATION STUDIES

**Decoder ablation** is shown in Table 2 (top). We keep all hyperparameters at defaults described in Section 3.2 and vary the number of decoder blocks (0 blocks corresponds to a simple linear projection) and compare to the ProtoRes baseline with 3 decoder blocks. We see consistent gain adding more decoder blocks at small compute cost (91, 95, 102, 106 hours of train time on miniMixamo dataset and NVIDIA M40 GPU for 0,1,2,3 blocks). Please see more detailed results in Appendix H.

**Prototype-Subtract-Accumulate ablation** is presented in Appendix I, in which we compare it against the ResPointNet stacking scheme (Maxpool-Concat daisy chain by Niemeyer et al. (2019)). We show that our stacking scheme is more accurate in a computationally efficient configuration with 3 encoder blocks and allows stacking deeper networks gaining more accuracy.

**GPD ablation** is shown in Table 2 (middle). We remove GPD and increase IKD depth to 6 blocks to match the capacity of IKD+GPD. Comparing to the baseline, we see that GPD creates consistent gain across metrics and datasets by conditioning IKD with a draft pose.

Table 2: Ablation studies on the random benchmark. Lower values are better.

| | | miniMixamo | | | miniUnity | | |
|---|---|---|---|---|---|---|---|
| | | $\mathcal{L}_{gpd-L2}^{det}$ | $\mathcal{L}_{ikd-L2}^{det}$ | $\mathcal{L}_{loc-geo}^{det}$ | $\mathcal{L}_{gpd-L2}^{det}$ | $\mathcal{L}_{ikd-L2}^{det}$ | $\mathcal{L}_{loc-geo}^{det}$ |
| | | ProtoRes baseline | | | | | |
| | | **1.36e-3** | **4.16e-3** | **0.2381** | **0.93e-3** | **3.28e-3** | **0.1817** |
| Decoder | blocks | Ablation of decoder | | | | | |
| | 0 | 1.54e-3 | 4.59e-3 | 0.2485 | 1.05e-3 | 3.65e-3 | 0.1939 |
| | 1 | 1.35e-3 | 4.34e-3 | 0.2433 | 0.93e-3 | 3.52e-3 | 0.1895 |
| | 2 | 1.34e-3 | 4.24e-3 | 0.2397 | 0.93e-3 | 3.34e-3 | 0.1840 |
| GPD | blocks, GPD/IKD | Ablation of GPD | | | | | |
| ✗ | 0/6 | 1.43e-3 | 4.39e-3 | 0.2413 | 0.93e-3 | 3.34e-3 | 0.1830 |
| $\mathcal{L}_{ikd-L2}^{*}$ | $\mathcal{L}_{loc-geo}^{*}$ | Ablation of rotation and position loss terms | | | | | |
| ✓ | ✗ | 1.60e-3 | 4.49e-3 | 0.2742 | 1.12e-3 | 3.65e-3 | 0.2392 |
| ✗ | ✓ | 2.04e-3 | 6.19e-3 | 0.2442 | 1.33e-3 | 4.63e-3 | 0.1862 |
| $W_{pos}$ | Randomized Loss | Ablation of randomized loss weighting | | | | | |
| 100 | ✗ | 1.77e-3 | 4.93e-3 | 0.2549 | 1.15e-3 | 3.58e-3 | 0.1905 |
| 1000 | ✗ | 1.66e-3 | 4.75e-3 | 0.2668 | 1.09e-3 | 3.40e-3 | 0.2029 |

**Ablation of loss terms,** shown in Table 2 (middle), studies the effect of (i) removing all L2 loss terms from the output of the FK pass and (ii) removing all Geodesic loss terms from the rotation output of IKD. Interestingly, removing either of the loss terms results in the degradation of all monitored metrics on both datasets. We conclude that jointly penalizing positions with L2 and rotations with Geodesic results in positive synergetic effects and improves the overall quality of pose model.

**Randomized loss weighting ablation** is shown in Table 2 (bottom). The randomized loss weighting scheme (see Algorithm 1 in Appendix C) is disabled by replacing all randomized loss terms with their deterministic counterparts. For example, $\mathcal{L}_{ikd-L2}^{rnd}$ is replaced with $\sum_{j=1}^{J} \text{MSE}(\mathbf{g}_j, \widehat{\mathbf{g}}_j)$. The inclusion of randomized weighting significantly improves generalization performance on all datasets and metrics. Additionally, when L2 weight $W_{pos}$ increases with disabled randomized weighting, position L2 metrics improve, but at the expense of declining rotation metrics. Therefore, randomized weighting scheme contributes positive effect that cannot be achieved by tweaking the deterministic loss weights.

**The limitations of the current work**, discussed in detail in Appendix M, include (i) the lack of temporal consistency as we focus on the problem of authoring a discrete pose, (ii) constraints are satisfied approximately, as opposed to the more conventional systems, (iii) exotic poses significantly deviating from the training data distribution may be hard to achieve.

## 4 CONCLUSIONS

We define and solve the discrete full-body pose authoring task using sparse and variable user inputs. We define and release two datasets to support the development of ML models for discrete pose authoring and animation. We propose ProtoRes, a novel ML architecture which processes a variable number of heterogeneous user inputs (position, angle, direction) to reconstruct a full-body pose. We compare ProtoRes against two strong ML baselines, Masked-FCR and Transformer, showing superior results for ProtoRes, both in terms of accuracy and computational efficiency. We also show that ML models reconstruct full-body poses from sparse user inputs more accurately than existing non-learnable inverse kinematics models. We develop a suite of UI tools for the integration of our model in Unity and provide demos showing how our model can be used effectively to solve the discrete pose authoring problem by the end user. Our results have a few implications. First, our ML based tools will have positive impacts on the simplification and democratization of the game development process by helping a wide audience materialize their creative animation ideas. Second, our novel approach to neural pose representation could be applied in a variety of tasks where efficient and accurate reconstruction of full-body poses from noisy intermittent measurements is important.

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

# Supplementary Material for ProtoRes: Proto-Residual Network for Pose Authoring via Learned Inverse Kinematics

## Table of Contents

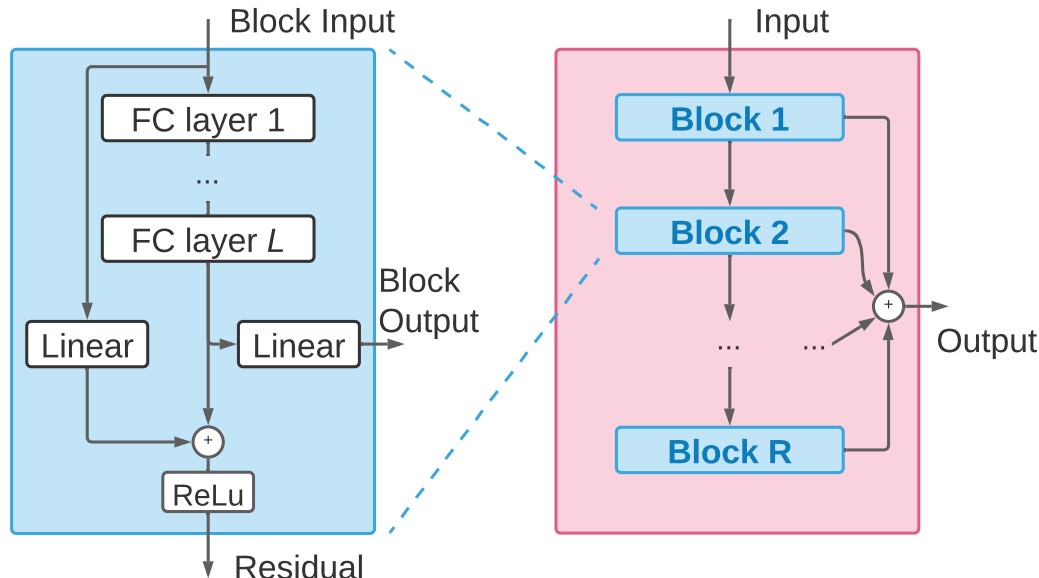

Figure 4: Block diagram of the fully-connected residual (FCR) decoder architecture. Left: the diagram of one residual block of the FCR decoder. Note that the basic residual block of the encoder architecture is exactly the same. Right: residual blocks connected in the FCR architecture.

## A  ARCHITECTURE DETAILS

**Residual block** depicted in Fig. 4 (left) is used as the basic building block of the ProtoRes encoder and decoder.

**Decoders** The block diagram of the global position and the inverse kinematics decoders used in the main architecture (see Fig. 2) is presented in Fig. 4. The architecture has fully connected residual topology consisting of multiple fully connected blocks connected using residual connections. Each block has residual and forward outputs. The forward output contributes to the final output of the decoder. The residual connection sums the hidden state of the block with the linear projection of the input and applies a ReLU non-linearity.

In the main text we use a convention that the number of layers and blocks in the encoder, as well as in GPD and IKD decoders is the same and is given by $L$ and $R$ respectively. Obviously, using a different number of layers and residual blocks in each of the blocks might be more optimal.

**Forward Kinematics** pass is applied to the output of the IKD, transforming local joint rotations and global root position into the global joint rotations and positions using skeleton kinematic equations. The FK pass relies on the offset vector $\mathbf{o}_j = [o_{x,j}, o_{y,j}, o_{z,j}]^\mathsf{T}$ and the rotation matrix $\mathbf{R}_j$ for each joint $j$. The offset vector is a fixed non-learnable vector representing bone length constraint for joint $j$. It provides the displacement of this joint with respect to its parent joint when joint $j$ rotation is zero. $\mathbf{R}_j$ can be naïvely represented using local Euler rotation angles $\alpha_j, \beta_j, \gamma_j$:

$$
\mathbf{o}_j = \begin{bmatrix} o_{x,j} \\ o_{y,j} \\ o_{z,j} \end{bmatrix}; \quad \mathbf{R}_j = \begin{bmatrix} \cos\alpha_j & -\sin\alpha_j & 0 \\ \sin\alpha_j & \cos\alpha_j & 0 \\ 0 & 0 & 1 \end{bmatrix} \begin{bmatrix} \cos\beta_j & 0 & \sin\beta_j \\ 0 & 1 & 0 \\ -\sin\beta_j & 0 & \cos\beta_j \end{bmatrix} \begin{bmatrix} 1 & 0 & 0 \\ 0 & \cos\gamma_j & -\sin\gamma_j \\ 0 & \sin\gamma_j & \cos\gamma_j \end{bmatrix}. \quad (13)
$$

However, we use a more robust representation proposed by (Zhou et al., 2019), relying on vector norm $\overrightarrow{\mathbf{u}} \equiv \mathbf{u}/\|\mathbf{u}\|_2$ and vector cross product $\mathbf{u} \times \mathbf{v} = \|\mathbf{u}\|\|\mathbf{v}\|\cos(\gamma)\overrightarrow{\mathbf{n}}$ ($\gamma$ is the angle between $\mathbf{u}$ and $\mathbf{v}$ in the plane containing them and $\overrightarrow{\mathbf{n}}$ is the normal to the plane):

$$
\widehat{\mathbf{r}}_{j,x} = \overrightarrow{\widehat{\mathbf{f}}_{R,j}[1:3]}, \quad \widehat{\mathbf{r}}_{j,z} = \overrightarrow{\widehat{\mathbf{r}}_{j,x} \times \widehat{\mathbf{f}}_{R,j}[4:6]}, \quad \widehat{\mathbf{r}}_{j,y} = \widehat{\mathbf{r}}_{j,z} \times \widehat{\mathbf{r}}_{j,x}, \quad \widehat{\mathbf{R}}_j = [\widehat{\mathbf{r}}_{j,x}\, \widehat{\mathbf{r}}_{j,y}\, \widehat{\mathbf{r}}_{j,z}]. \quad (14)
$$

Provided with the local offset vectors and rotation matrices of all joints, the global rigid transform of any joint $j$ is predicted following the tree recursion from the parent joint $p(j)$ of joint $j$:

$$\widehat{\mathbf{G}}_j = \widehat{\mathbf{G}}_{p(j)} \begin{bmatrix} \widehat{\mathbf{R}}_j & \mathbf{o}_j \\ \mathbf{0} & 1 \end{bmatrix}. \tag{15}$$

The global transform matrix $\widehat{\mathbf{G}}_j$ of joint $j$ contains its global rotation matrix, $\widehat{\mathbf{G}}_j^{13} \equiv \widehat{\mathbf{G}}_j[1:3,1:3]$, and its 3D global position, $\widehat{\mathbf{g}}_j = \widehat{\mathbf{G}}_j[1:3,4]$.

## B  EFFECTOR NOISE MODEL

This section describes the details of the NOISEMODEL that is used in Algorithm 1 to corrupt model effector input $\mathbf{x}[i,:]$ based on appropriate noise level $\sigma(\Lambda_i)$.

### B.1  POSITION EFFECTOR NOISE MODEL

If effector type is positional ($T_i = 0$), *i.e.* effector $i$ is a coordinate in 3D space, typically corresponding to the desired position of joint $I_i$ in 3D space, we employ Gaussian white noise model:

$$\mathbf{x}[i,1:3] = \mathbf{g}_{I_i} + \sigma(\Lambda_i)\varepsilon_i; \quad \mathbf{x}[i,4:6] = 0. \tag{16}$$

Here $\mathbf{x}[i,:]$ is the $i$-th model input, $\mathbf{g}_{I_i}$ is the ground truth location of joint $I_i$, $\sigma(\Lambda_i)$ is the noise standard deviation computed based on eq. (20) and $\varepsilon_i$ is a 3D vector sampled from the zero-mean Normal distribution $\mathcal{N}(\mathbf{0},\mathbf{I})$.

### B.2  ROTATION EFFECTOR NOISE MODEL

If effector type is angular ($T_i = 1$), *i.e.* effector $i$ is a 6DoF rotation matrix representation, we employ random rotation model that is implemented in the following stages. First, suppose $\mathbf{f}_{I_i}$ is the ground truth 6DoF representation of the global rotation of joint $I_i$ corresponding to effector $i$. We transform it to the rotation matrix representation $\mathbf{G}_{I_i}^{13}$ using equation (14). Second, we generate the random 3D Euler angles vector $\varepsilon_i$ from the zero-mean Gaussian distribution $\mathcal{N}(\mathbf{0}, \sigma(\Lambda_i)\mathbf{I})^1$ and convert it to the random rotation matrix $\Psi_i$ using eq. (13):

$$\Psi_i = \begin{bmatrix} \cos\varepsilon_i[1] & -\sin\varepsilon_i[1] & 0 \\ \sin\varepsilon_i[1] & \cos\varepsilon_i[1] & 0 \\ 0 & 0 & 1 \end{bmatrix} \begin{bmatrix} \cos\varepsilon_i[2] & 0 & \sin\varepsilon_i[2] \\ 0 & 1 & 0 \\ -\sin\varepsilon_i[2] & 0 & \cos\varepsilon_i[2] \end{bmatrix} \begin{bmatrix} 1 & 0 & 0 \\ 0 & \cos\varepsilon_i[3] & -\sin\varepsilon_i[3] \\ 0 & \sin\varepsilon_i[3] & \cos\varepsilon_i[3] \end{bmatrix}. \tag{17}$$

Third, we apply random rotation to the ground truth matrix, $\mathbf{G}_{I_i}^{13\prime} = \Psi_i \mathbf{G}_{I_i}^{13}$. Finally, we convert the randomly perturbed rotation matrix back to the 6DoF representation:

$$\mathbf{x}[i,1:3] = \mathbf{G}_{I_i}^{13\prime}[:,1], \quad \mathbf{x}[i,4:6] = \mathbf{G}_{I_i}^{13\prime}[:,2]. \tag{18}$$

### B.3  LOOK-AT EFFECTOR NOISE MODEL

If effector type is look-at ($T_i = 2$), *i.e.* effector $i$ is a position of the target at which a given joint is supposed to look, we employ random sampling of the target point along the ray cast in the direction formed by the global rotation of a given joint.

First, we sample the local direction vector $\mathbf{d}_i$ from the zero-mean normal 3D distribution $\mathcal{N}(\mathbf{0},\mathbf{I})$ and normalize it to unit length. Second, we sample the distance between the joint and the target object, $d_\mathbf{t}$, from the normal distribution $\mathcal{N}(0,5)$ folded over at 0 by taking the absolute value. The location of the target object is then determined as $\mathbf{t}_i = \mathbf{g}_{I_i} + d_\mathbf{t}\mathbf{d}_i + \sigma(\Lambda_i)\varepsilon_i$. Finally, the output is constructed as follows:

$$\mathbf{x}[i,1:3] = \mathbf{t}_i, \quad \mathbf{x}[i,4:6] = \mathbf{d}_i. \tag{19}$$

As previously, $\varepsilon_i$ is a 3D vector sampled from the zero-mean Normal distribution $\mathcal{N}(\mathbf{0},\mathbf{I})$.

---

**Algorithm 1** Loss calculation for a single item in the training batch of ProtoRes.

---

**Require:** $\mathbf{R}_j, \mathbf{G}_j; N \sim \text{UNIFORM}[3, 16]$  ▷ Ground truth for all joints $j \in [0, J)$; number of effectors
**Ensure: x**  ▷ Sample inputs
  $I_1, \ldots, I_N \leftarrow \text{MULTINOMIAL}(\{0, \ldots, J-1\}, N)$  ▷ Effector IDs
  $T_1, \ldots, T_N \leftarrow \text{MULTINOMIAL}(\{0, 1, 2\}, N)$  ▷ Effector type
  **for** $i$ in $1 \ldots N$ **do**
    $\Lambda_i \leftarrow \text{UNIFORM}[0, 1]$  ▷ Effector tolerance
    $\sigma(\Lambda_i); W(\Lambda_i) \leftarrow \sigma_M \Lambda_i^\eta; \min(W_M, 1/\sigma(\Lambda_i))$  ▷ Effector noise std and weight
    $\mathbf{x}[i, :] \leftarrow \text{NOISEMODEL}(\mathbf{G}_{I_i}, \sigma(\Lambda_i), T_i)$  ▷ Generate noisy effector
  **end for**
**Predict:** $\widetilde{\mathbf{f}}_{R,j}, \widehat{\mathbf{R}}_j, \widehat{\mathbf{G}}_j \quad \forall j$ based on $\mathbf{x}$
  $\mathcal{L}_{gpd-L2}^{rnd} \leftarrow \frac{1}{\sum_{i=1}^N \mathbb{1}_{T_i=0} W(\Lambda_i)} \sum_{i=1}^N \mathbb{1}_{T_i=0} W(\Lambda_i) \text{MSE}(\mathbf{g}_{I_i}, \widetilde{\mathbf{f}}_{R,I_i})$  ▷ Randomized GPD position loss
  $\mathcal{L}_{ikd-L2}^{rnd} \leftarrow \frac{1}{\sum_{i=1}^N \mathbb{1}_{T_i=0} W(\Lambda_i)} \sum_{i=1}^N \mathbb{1}_{T_i=0} W(\Lambda_i) \text{MSE}(\mathbf{g}_{I_i}, \widehat{\mathbf{g}}_{I_i})$  ▷ Randomized IKD position loss
  $\mathcal{L}_{gpd-L2}^{det} \leftarrow \sum_{j=1}^J \text{MSE}(\mathbf{g}_j, \widetilde{\mathbf{f}}_{R,j})$  ▷ Deterministic GPD position loss
  $\mathcal{L}_{ikd-L2}^{det} \leftarrow \sum_{j=1}^J \text{MSE}(\mathbf{g}_j, \widehat{\mathbf{g}}_j)$  ▷ Deterministic IKD position loss
  $\mathcal{L}_{loc-geo}^{det} \leftarrow \sum_{j=1}^J \text{GEO}(\mathbf{R}_j, \widehat{\mathbf{R}}_j)$  ▷ Deterministic local rotation loss
  $\mathcal{L}_{glob-geo}^{rnd} \leftarrow \frac{1}{\sum_{i=1}^N \mathbb{1}_{T_i=1} W(\Lambda_i)} \sum_{i=1}^N \mathbb{1}_{T_i=1} W(\Lambda_i) \text{GEO}(\mathbf{G}_{I_i}^{13}, \widehat{\mathbf{G}}_{I_i}^{13})$  ▷ Randomized global rotation loss
  $\mathcal{L}_{lat}^{det} \leftarrow \frac{1}{\sum_{i=1}^N \mathbb{1}_{T_i=2}} \sum_{i=1}^N \mathbb{1}_{T_i=2} \text{LAT}(\mathbf{x}[i, 1:3], \mathbf{x}[i, 4:6], \widehat{\mathbf{G}}_{I_i}^{13})$  ▷ Randomized Look-at loss
  $\mathcal{L} \leftarrow \frac{W_{pos}}{J}(\mathcal{L}_{gpd-L2}^{rnd} + \mathcal{L}_{ikd-L2}^{rnd} + \mathcal{L}_{gpd-L2}^{det} + \mathcal{L}_{ikd-L2}^{det}) + \frac{1}{J}(\mathcal{L}_{lat}^{det} + \mathcal{L}_{glob-geo}^{rnd} + \mathcal{L}_{loc-geo}^{det})$  ▷ Total loss

---

## C   TRAINING AND EVALUATION METHODOLOGY: DETAILS

The training methodology involves techniques targeting to (i) regularize model via data augmentation, (ii) learn handling of sparse inputs and (iii) effectively combine multi-task loss terms.

**Data augmentation** is based on the rotation and mirror augmentations. The former rotates the skeleton around the vertical $Y$ axis by a random angle in $[0, 2\pi]$. Rotation w.r.t. ground $XZ$ plane is not applied to avoid creating poses implausible according to the gravity direction. Mirror augmentation removes any implicit left- or right-handedness biases by flipping the skeleton w.r.t. the $YZ$ plane.

**Sparse inputs** modeling relies on effector sampling. First, the total number of effectors is sampled uniformly at random in the range [3, 16]. Given the total number of effectors, the effector IDs (one of 64 joints) and types (one of 3 types: position, rotation, or look-at) are sampled from the Multinomial without replacement. This sampling scheme produces an exponentially large number of different permutations of effector types and joints, resulting in strong regularizing effects.

**Effector tolerance and randomized loss weighting.** The motivation behind the randomized loss weighting is two-fold. First, the randomized loss weighting was originally introduced as a binary indicator to force the model to better respect constraints provided as effectors, compared to the joints predicted by the model. Afterwards, we realized that this can be made more flexible by generating a continuous variable representing the tolerance level. This variable can be provided as an input to the network and it can be exposed as a user interface feature to let the user control the degree of responsiveness of the model to different effectors. We also discovered that the latter feature only works when a noise is added to effector value and the standard deviation of the noise is appropriately synchronised with the tolerance. The noise teaches the model to disregard the effector completely if the tolerance input value corresponds to the high noise variance regime.

Second, we observed that the use of the randomized weighting improves multi-task training and generalization performance. Initially, we noticed that increasing the weight of position loss would drive the generalization on the position metric to a better spot, while the rotation metric generalization would be compromised, which is not surprising. This was especially evident when the position loss weight was increased by one or two orders of magnitude. This is a well-known phenomenon when

---

[1]Note that in the case of angles, sampling from the Tikhonov (a.k.a. circular normal or von Mises) distribution might be a better idea, but Gaussian worked well in our case.

dealing with multiple loss terms, which we informally call "fighting" between losses (related to the Pareto front, more formally). This effect can be observed when comparing two bottom rows in Table 2. Introducing the randomized loss weighting scheme we observed two things. "Fighting" disappeared, i.e. the randomly generated weights of position effectors varied in a wide range between 1e-1 and 1e5 within a batch, but the fact that some of the weight values are one or two orders of magnitude greater than the baseline position weight of 100, did not lead to the deterioration of the rotation loss. Moreover, the introduction of the randomized loss weighting positively affected the generalization on both position and rotation metrics, which can be assessed by comparing the first row of Table 2 with its bottom rows. This leads us to believe that the randomized loss weighting introduces a sinergy in the multi-task training that is not achievable by simple adjustment of static loss weights. We believe this technique could be more generally applicable to multi-task training, but a more detailed investigation of this is outside of the current scope.

We now describe the technical details behind randomized loss weighting implementation. For each sampled effector, we further uniformly sample $\Lambda \in [0,1]$ treated as effector tolerance. Given an effector tolerance $\Lambda$, noise (noise models used for different effector types are described in detail in Appendix B) with variance proportional to $\Lambda$ is added to effector data before feeding them to the neural network:

$$\sigma(\Lambda) = \sigma_M \Lambda^\eta. \tag{20}$$

We use $\eta > 10$ to shape the distribution of $\sigma$ to smaller values. Furthermore, to each effector is attached a randomized loss weight reciprocal to $\sigma(\Lambda)$, capped at $W_M$ if $\sigma(\Lambda) < 1/W_M$:

$$W(\Lambda) = \min(W_M, 1/\sigma(\Lambda)). \tag{21}$$

$\Lambda$ drives network inputs and is simultaneously used to weigh losses by $W(\Lambda)$. Thus ProtoRes learns to respect effector tolerance, leading to two positive outcomes. First, ProtoRes provides a tool allowing one to emphasize small tolerance effectors ($\Lambda \approx 0$) and relax the large tolerance ones ($\Lambda \approx 1$). Second, randomized loss weighting improves the overall accuracy in the multi-task training scenario.

The detailed procedure to compute the ProtoRes loss based on one batch item is presented in Algorithm 1 and the summary is provided below. First, we sample (i) the number of effectors and (ii) their associated type and ID. For each effector, we randomly sample the tolerance level and compute the associated noise std and loss weight. Given noise std, an appropriate noise model is applied to generate input data based on effector type as described in Appendix B. Then ProtoRes predicts draft joint positions $\widetilde{\mathbf{f}}_{R,j}$, local joint rotations $\widehat{\mathbf{R}}_j$, as well as world-space rotations and positions $\widehat{\mathbf{G}}_j$ for all joints $j \in [0, J)$. We conclude by calculating the individual deterministic and randomized loss terms, whose weighted sum is used for backpropagation.

# D  DATASETS: DETAILS

## D.1  DATASETS DESCRIPTIONS

**miniMixamo**  We use the following procedure to create our first dataset from the publicly available MOCAP data available from `mixamo.com`, generously provided by Adobe Inc. (2020). We download a total of 1598 clips and retarget them on our custom 64-joint skeleton using the Mixamo online tool. This skeleton definition is used in Unity to extract the global positions as well as global and local rotations of each joint at the rate of 60 frames per second (total 356,545 frames). The resulting dataset is partitioned at the clip level into train/validation/test splits (with proportion 0.8/0.1/0.1, respectively) by sampling clip IDs uniformly at random. Splitting by clip makes the evaluation framework more realistic and less prone to overfitting: frames belonging to the same clip are often similar. At last, the final splits retain only 10% of randomly sampled frames (miniMixamo has 33,676 frames total after subsampling) and all the clip identification information (clip ID, meta-data/description, character information, etc.) is discarded. This anonymization guarantees that the original sequences from `mixamo.com` cannot be reconstructed from our dataset, allowing us to release the dataset for reproducibility purposes without violating the original dataset license (Adobe Inc., 2020).

For miniMixamo our contribution is as follows. Mixamo data is not available as a single file. Therefore, anyone who wants to use the data for academic purposes needs to go through a lengthy process of downloading individual files. Importantly, this step creates additional risks for the reproducibility

of results. We have gone through this step and assembled all files in one place. Furthermore, Mixamo data cannot be redistributed, according to Adobe licensing, which is again a reproducibility risk. However, we do not need the entire dataset for benchmarking on the task we defined. Therefore, we defined a suitable subsampling and anonymization procedure that allowed us to obtain (i) a high quality reproducible benchmark dataset for our task and (ii) a legal permission from Mixamo/Adobe to redistribute this benchmark for academic research purposes. We are extremely grateful to the representatives from Mixamo and Adobe who approved it to facilitate the democratization of character animation. The entire process of creating the benchmark took us a few months of work, which we consider a significant contribution to the research community.

**miniUnity**   To collect our second dataset we predefine a wide range of human motion scenarios and hire a qualified MOCAP studio to record 1776 clips (967,258 total frames @60 fps). Then we create a dataset of a total of 96,666 subsampled frames following exactly the same methodology that was employed for miniMixamo.

The following action scenarios were used to collect MOCAP sequences in miniUnity. The detailed hierarchy of motion scenarios is presented in Table 6. First, the following locomotion types: compass crouch, compass jogs, compass runs, compass walks were collected for female and male subjects under high energy, low energy and injured scenarios. The same locomotion types were collected under neutral energy feminine and neutral energy macho scenarios. Furthermore, under neutral energy generic scenario, for both female and male subjects, we collected following action, object and environment interaction types: archery, bokken fighting, calisthetics, door interactions, fist fighting, food, handgun, hands, knife fighting, locomotion, longsword fighting, phone, place, railing interactions, rifle, seated interactions, shotgun, standings, sword, wall. Among the latter categories, locomotion and handgun had following more detailed subdivisions. Locomotion: compass crawls, compass crouch, compass jogs, compass rifle aim walks, compass rifle crawl, compass rifle crouch, compass rifle jogs, compass rifle runs, compass rifle walks, compass runs, compass walks, rifle idles, walk carry heavy backpack, walk carry heavy sack, walk carry ladder, walk dragging heavy object. Handgun: downwards, level, upwards, verticals. The motion categories existing in miniUnity and miniMixamo can be compared by looking at Tables 6 and 7 respectively.

The key differentiators of the datasets that we release that make them significant contributions toward AI driven artistic pose development are as follows:

- Both miniMixamo (derived from the Mixamo, which is generously provided by Adobe) and miniUnity are collected by professional studio contractors relying on the service of professional actors using high-end MOCAP studio equipment.

- Both datasets are clean and contain data of very high quality. For our dataset, we specifically had to go through multiple cleaning iterations to make sure all the data collection and conversion artifacts are removed. We are very grateful to our contractor for being diligent, detail oriented, and determined to provide the high quality data.

- Both datasets provide data in the industry standard skeleton format compatible with multiple existing animation rigs and therefore making it easy to experiment with the ML assisted pose authoring results in 3D development environments such as Unity. This is in contrast to CMU and AMASS datasets that are collected in heterogeneous environments using non-standard sensor placements.

- Both our datasets provide 64 joint skeletons and contain fine grain hands and feet data, unlike other publicly available datasets.

## D.2   DATASETS DESCRIPTIVE STATISTICS

The standard deviations of joint positions (in the coordinate system relative to hips joint) and joint local quaternions are presented in Tables 3 and 4.

The distributions of 30 most popular tags across the frames of the original dataset used to build miniMixamo and miniUnity are shown in Figure 5. It is clear that the two dataset cover some common activities such as walking and idling, for example. Additionally, there are numerous categories the two datasets emphasize separately. For example miniMixamo focuses a lot on fighting and handling guns,

whereas miniUnity has more neutral energy activities, provides extensive labeling of male/female poses as well as covers scenarios of handling objects and food.

The distribution of PCA of flattened rotations of all joints of miniMixamo (red) and miniUnity (blue) datasets is shown in Figure 6. Each point corresponds to a pose. It is clear that there are both overlapping areas corresponding to clusters with similar poses and areas with disjoint clusters, showing distinct pose configurations characteristic of the two different datasets.

Table 3: Per-joint hip-local positions standard deviations computed over two proposed datasets

| Joint | miniMixamo | | | miniAnonymous | | |
|---|---|---|---|---|---|---|
| | X | Y | Z | X | Y | Z |
| Hips | 0 | 0 | 0 | 0 | 0 | 0 |
| Spine0 | 0 | 0 | 0 | 0 | 0 | 0 |
| Spine1 | 0.0049 | 0.0034 | 0.0125 | 0.0019 | 0.0003 | 0.0045 |
| Chest | 0.0153 | 0.0114 | 0.0353 | 0.008 | 0.0032 | 0.0238 |
| Neck | 0.0395 | 0.0338 | 0.0859 | 0.0257 | 0.0245 | 0.0805 |
| Head | 0.066 | 0.0609 | 0.1288 | 0.0484 | 0.0608 | 0.1175 |
| ClavicleLeft | 0.0409 | 0.0481 | 0.0798 | 0.0254 | 0.0414 | 0.0743 |
| ClavicleRight | 0.0412 | 0.0475 | 0.0794 | 0.0255 | 0.0412 | 0.0741 |
| BicepLeft | 0.0469 | 0.0723 | 0.0971 | 0.0297 | 0.0483 | 0.093 |
| ForarmLeft | 0.0941 | 0.1608 | 0.1577 | 0.0825 | 0.1293 | 0.1563 |
| HandLeft | 0.174 | 0.2697 | 0.1948 | 0.1564 | 0.2381 | 0.1964 |
| Index0Left | 0.2026 | 0.3167 | 0.21 | 0.1871 | 0.2878 | 0.2116 |
| Index1Left | 0.2121 | 0.3312 | 0.2164 | 0.1978 | 0.3022 | 0.2183 |
| Index2Left | 0.2184 | 0.3415 | 0.2208 | 0.2068 | 0.3133 | 0.2238 |
| Index2LeftEnd | 0.2247 | 0.3519 | 0.2259 | 0.2151 | 0.324 | 0.2292 |
| Middle0Left | 0.2049 | 0.317 | 0.2147 | 0.1862 | 0.2873 | 0.2161 |
| Middle1Left | 0.2155 | 0.3333 | 0.2216 | 0.1979 | 0.3028 | 0.2235 |
| Middle2Left | 0.2218 | 0.3437 | 0.226 | 0.2069 | 0.3144 | 0.2294 |
| Middle2LeftEnd | 0.2278 | 0.354 | 0.2314 | 0.2151 | 0.3258 | 0.2351 |
| Ring0Left | 0.2081 | 0.318 | 0.2207 | 0.1866 | 0.2872 | 0.2221 |
| Ring1Left | 0.2175 | 0.3324 | 0.2268 | 0.1974 | 0.3017 | 0.2296 |
| Ring2Left | 0.2225 | 0.341 | 0.2304 | 0.2053 | 0.3126 | 0.2352 |
| Ring2LeftEnd | 0.2273 | 0.3493 | 0.2349 | 0.2129 | 0.3231 | 0.2408 |
| Pinky0Left | 0.2106 | 0.3178 | 0.2262 | 0.187 | 0.2859 | 0.2276 |
| Pinky1Left | 0.2187 | 0.3301 | 0.2315 | 0.1969 | 0.2982 | 0.235 |
| Pinky2Left | 0.2232 | 0.3379 | 0.2347 | 0.2041 | 0.3071 | 0.2402 |
| Pinky2LeftEnd | 0.2272 | 0.3449 | 0.2382 | 0.2095 | 0.3142 | 0.2449 |
| Thumb0Left | 0.1884 | 0.2971 | 0.1982 | 0.1757 | 0.2681 | 0.2003 |
| Thumb1Left | 0.197 | 0.3108 | 0.2039 | 0.1849 | 0.2791 | 0.2021 |
| Thumb2Left | 0.2073 | 0.3258 | 0.2121 | 0.1961 | 0.2935 | 0.2075 |
| Thumb2LeftEnd | 0.2162 | 0.3374 | 0.2195 | 0.2054 | 0.3054 | 0.2124 |
| BicepRight | 0.05 | 0.0725 | 0.0969 | 0.0296 | 0.0543 | 0.086 |
| ForarmRight | 0.098 | 0.17 | 0.1524 | 0.0897 | 0.138 | 0.1603 |
| HandRight | 0.1703 | 0.2842 | 0.185 | 0.169 | 0.2372 | 0.1899 |
| Index0Right | 0.196 | 0.3329 | 0.2045 | 0.2037 | 0.2843 | 0.2093 |
| Index1Right | 0.2048 | 0.348 | 0.2118 | 0.215 | 0.2988 | 0.2168 |
| Index2Right | 0.2099 | 0.3586 | 0.2173 | 0.2251 | 0.3114 | 0.2223 |
| Index2RightEnd | 0.2145 | 0.3691 | 0.2234 | 0.2352 | 0.3235 | 0.227 |
| Middle0Right | 0.1984 | 0.3335 | 0.2077 | 0.2025 | 0.2821 | 0.2123 |
| Middle1Right | 0.2078 | 0.35 | 0.2154 | 0.2147 | 0.2968 | 0.2194 |
| Middle2Right | 0.2119 | 0.3586 | 0.2187 | 0.2221 | 0.3061 | 0.2232 |
| Middle2RightEnd | 0.215 | 0.3649 | 0.2207 | 0.2274 | 0.3133 | 0.2258 |
| Ring0Right | 0.2017 | 0.3347 | 0.212 | 0.2021 | 0.2803 | 0.2165 |
| Ring1Right | 0.2099 | 0.3495 | 0.2187 | 0.2142 | 0.2946 | 0.224 |
| Ring2Right | 0.2133 | 0.3566 | 0.2218 | 0.223 | 0.3047 | 0.2284 |
| Ring2RightEnd | 0.2153 | 0.3612 | 0.2234 | 0.2304 | 0.3136 | 0.2322 |
| Pinky0Right | 0.2043 | 0.3346 | 0.2158 | 0.2014 | 0.278 | 0.2201 |
| Pinky1Right | 0.2112 | 0.3472 | 0.2215 | 0.2118 | 0.2888 | 0.2261 |
| Pinky2Right | 0.2144 | 0.3544 | 0.2251 | 0.2192 | 0.2969 | 0.2296 |
| Pinky2RightEnd | 0.2165 | 0.3591 | 0.2276 | 0.2242 | 0.3039 | 0.2316 |
| Thumb0Right | 0.1828 | 0.3126 | 0.1926 | 0.1911 | 0.2676 | 0.198 |
| Thumb1Right | 0.1907 | 0.3265 | 0.1985 | 0.2013 | 0.2801 | 0.2021 |
| Thumb2Right | 0.1996 | 0.3399 | 0.205 | 0.2123 | 0.2949 | 0.2091 |
| Thumb2RightEnd | 0.2064 | 0.3494 | 0.2096 | 0.2205 | 0.3061 | 0.2148 |
| ThighLeft | 0 | 0 | 0 | 0 | 0 | 0 |
| CalfLeft | 0.083 | 0.1416 | 0.1341 | 0.0673 | 0.152 | 0.1385 |
| FootLeft | 0.1263 | 0.1611 | 0.1942 | 0.1185 | 0.1861 | 0.2062 |
| ToeLeft | 0.1453 | 0.1705 | 0.2202 | 0.1338 | 0.1943 | 0.2308 |
| ToeLeftEnd | 0.1632 | 0.1789 | 0.2254 | 0.1494 | 0.1985 | 0.2349 |
| ThighRight | 0 | 0 | 0 | 0 | 0 | 0 |
| CalfRight | 0.0781 | 0.1352 | 0.1381 | 0.0695 | 0.1495 | 0.1436 |
| FootRight | 0.1258 | 0.1659 | 0.2047 | 0.1181 | 0.1908 | 0.209 |
| ToeRight | 0.1435 | 0.1743 | 0.2342 | 0.1319 | 0.1937 | 0.2353 |
| ToeRightEnd | 0.158 | 0.1788 | 0.2414 | 0.1471 | 0.1924 | 0.2402 |

Table 4: Per-joint quaternion component standard deviations computed over two proposed datasets

| Joint | miniMixamo | | | | miniAnonymous | | | |
|---|---|---|---|---|---|---|---|---|
| | X | Y | Z | W | X | Y | Z | W |
| Hips | 0.187 | 0.2239 | 0.0952 | 0.5381 | 0.1614 | 0.4735 | 0.0951 | 0.4673 |
| Spine0 | 0.0928 | 0.0343 | 0.0358 | 0.0133 | 0.0325 | 0.019 | 0.0139 | 0.0015 |
| Spine1 | 0.0579 | 0.038 | 0.0344 | 0.0076 | 0.0694 | 0.0308 | 0.021 | 0.0065 |
| Chest | 0.0612 | 0.0292 | 0.0227 | 0.0065 | 0.0695 | 0.0319 | 0.0204 | 0.0075 |
| Neck | 0.1241 | 0.0682 | 0.0572 | 0.0236 | 0.1368 | 0.0887 | 0.0691 | 0.0347 |
| Head | 0.1183 | 0.1206 | 0.0703 | 0.0374 | 0.1349 | 0.0948 | 0.0746 | 0.0337 |
| ClavicleLeft | 0.2139 | 0.2156 | 0.2234 | 0.2039 | 0.4424 | 0.3554 | 0.4045 | 0.3964 |
| ClavicleRight | 0.4307 | 0.4887 | 0.4985 | 0.4309 | 0.4785 | 0.459 | 0.475 | 0.4696 |
| BicepLeft | 0.188 | 0.2132 | 0.1935 | 0.0744 | 0.2015 | 0.2523 | 0.2225 | 0.1059 |
| ForarmLeft | 0.1131 | 0.1535 | 0.259 | 0.1768 | 0.1958 | 0.2931 | 0.2892 | 0.2372 |
| HandLeft | 0.1732 | 0.2283 | 0.1276 | 0.2091 | 0.0034 | 0.1179 | 0.0603 | |
| Index0Left | 0.188 | 0.0278 | 0.047 | 0.0623 | 0.2013 | 0.0117 | 0.063 | 0.055 |
| Index1Left | 0.222 | 0.0055 | 0.0225 | 0.1145 | 0.1915 | 0.0007 | 0.0014 | 0.0787 |
| Index2Left | 0.178 | 0.0056 | 0.0157 | 0.0675 | 0.1743 | 0.0006 | 0.0014 | 0.0587 |
| Index2LeftEnd | 0 | 0 | 0 | 0 | 0.0007 | 0.0006 | 0.0014 | 0.0013 |
| Middle0Left | 0.2031 | 0.0203 | 0.0395 | 0.0757 | 0.2614 | 0.0175 | 0.0166 | 0.0933 |
| Middle1Left | 0.2265 | 0.0056 | 0.0236 | 0.1177 | 0.1648 | 0.0033 | 0.0017 | 0.0594 |
| Middle2Left | 0.186 | 0.0052 | 0.0153 | 0.0714 | 0.1758 | 0.0012 | 0.0017 | 0.0522 |
| Middle2LeftEnd | 0 | 0 | 0 | 0 | 0.0007 | 0.0005 | 0.0014 | 0.0012 |
| Ring0Left | 0.2151 | 0.0242 | 0.046 | 0.0908 | 0.2561 | 0.0249 | 0.0297 | 0.087 |
| Ring1Left | 0.2306 | 0.0075 | 0.0235 | 0.1218 | 0.1867 | 0.0004 | 0.0014 | 0.0773 |
| Ring2Left | 0.186 | 0.0058 | 0.0179 | 0.0706 | 0.1498 | 0.0005 | 0.0014 | 0.048 |
| Ring2LeftEnd | 0 | 0 | 0 | 0 | 0.0007 | 0.0005 | 0.0014 | 0.0012 |
| Pinky0Left | 0.2179 | 0.0377 | 0.0701 | 0.094 | 0.2533 | 0.0585 | 0.0901 | 0.0874 |
| Pinky1Left | 0.2133 | 0.0104 | 0.0238 | 0.0929 | 0.1978 | 0.0004 | 0.0014 | 0.0853 |
| Pinky2Left | 0.1999 | 0.009 | 0.0187 | 0.1088 | 0.1798 | 0.0008 | 0.0012 | 0.0701 |
| Pinky2LeftEnd | 0 | 0 | 0 | 0 | 0.0009 | 0.0008 | 0.0012 | 0.0012 |
| Thumb0Left | 0.1266 | 0.0871 | 0.2072 | 0.0443 | 0.0957 | 0.1374 | 0.071 | 0.1124 |
| Thumb1Left | 0.0447 | 0.0613 | 0.0984 | 0.0422 | 0.0635 | 0.0006 | 0.0006 | 0.0127 |
| Thumb2Left | 0.0625 | 0.0502 | 0.1524 | 0.0531 | 0.1265 | 0.0004 | 0.0007 | 0.0174 |
| Thumb2LeftEnd | 0 | 0 | 0 | 0 | 0.0012 | 0.0004 | 0.0007 | 0.001 |
| BicepRight | 0.1912 | 0.206 | 0.2099 | 0.0897 | 0.2043 | 0.2555 | 0.2305 | 0.1083 |
| ForarmRight | 0.1039 | 0.1349 | 0.3054 | 0.2408 | 0.2122 | 0.2554 | 0.4638 | 0.4425 |
| HandRight | 0.1609 | 0.1848 | 0.1284 | 0.0902 | 0.2169 | 0.0082 | 0.15 | 0.0648 |
| Index0Right | 0.2545 | 0.0292 | 0.0448 | 0.9321 | 0.197 | 0.0148 | 0.0535 | 0.0535 |
| Index1Right | 0.3376 | 0.0053 | 0.0201 | 0.9063 | 0.1763 | 0.0013 | 0.0009 | 0.0662 |
| Index2Right | 0.2512 | 0.0055 | 0.0155 | 0.9351 | 0.158 | 0.0008 | 0.0013 | 0.0449 |
| Index2RightEnd | 0 | 0 | 0 | 0 | 0.0008 | 0.0009 | 0.0013 | 0.0014 |
| Middle0Right | 0.2965 | 0.0163 | 0.0516 | 0.9213 | 0.261 | 0.0177 | 0.0176 | 0.0915 |
| Middle1Right | 0.4421 | 0.0074 | 0.0226 | 0.8616 | 0.1968 | 0.0048 | 0.0017 | 0.0765 |
| Middle2Right | 0.2967 | 0.0079 | 0.0154 | 0.9217 | 0.2133 | 0.0014 | 0.0014 | 0.0762 |
| Middle2RightEnd | 0 | 0 | 0 | 0 | 0.0036 | 0.0009 | 0.0012 | 0.0013 |
| Ring0Right | 0.3277 | 0.026 | 0.0569 | 0.911 | 0.2494 | 0.0263 | 0.0405 | 0.0754 |
| Ring1Right | 0.4355 | 0.0079 | 0.0238 | 0.8645 | 0.2002 | 0.001 | 0.0011 | 0.073 |
| Ring2Right | 0.3114 | 0.0099 | 0.0167 | 0.9168 | 0.1543 | 0.0009 | 0.0012 | 0.0415 |
| Ring2RightEnd | 0 | 0 | 0 | 0 | 0.0009 | 0.0009 | 0.0012 | 0.0012 |
| Pinky0Right | 0.3372 | 0.0452 | 0.076 | 0.9051 | 0.2767 | 0.0574 | 0.0737 | 0.1006 |
| Pinky1Right | 0.3847 | 0.0131 | 0.0251 | 0.8887 | 0.1991 | 0.001 | 0.001 | 0.0786 |
| Pinky2Right | 0.314 | 0.0109 | 0.0171 | 0.9143 | 0.1973 | 0.001 | 0.0011 | 0.0668 |
| Pinky2RightEnd | 0 | 0 | 0 | 0 | 0.001 | 0.001 | 0.0011 | 0.0012 |
| Thumb0Right | 0.1855 | 0.0962 | 0.1144 | 0.9211 | 0.2739 | 0.6352 | 0.1285 | 0.4566 |
| Thumb1Right | 0.0484 | 0.0561 | 0.265 | 0.9293 | 0.0747 | 0.0007 | 0.0012 | 0.0179 |
| Thumb2Right | 0.0652 | 0.0359 | 0.2321 | 0.9364 | 0.169 | 0.0009 | 0.001 | 0.04 |
| Thumb2RightEnd | 0 | 0 | 0 | 0 | 0.0012 | 0.0009 | 0.001 | 0.0014 |
| ThighLeft | 0.107 | 0.0975 | 0.0818 | 0.2296 | 0.1164 | 0.0884 | 0.0704 | 0.2397 |
| CalfLeft | 0.2419 | 0.075 | 0.0448 | 0.145 | 0.2698 | 0.049 | 0.0427 | 0.1751 |
| FootLeft | 0.1158 | 0.0685 | 0.073 | 0.0554 | 0.1258 | 0.091 | 0.0894 | 0.0697 |
| ToeLeft | 0.0658 | 0.026 | 0.018 | 0.0552 | 0.052 | 0.045 | 0.0365 | 0.0328 |
| ToeLeftEnd | 0 | 0 | 0 | 0 | 0.0009 | 0.0002 | 0.0009 | 0.0008 |
| ThighRight | 0.1083 | 0.0959 | 0.0783 | 0.2298 | 0.1234 | 0.0895 | 0.0783 | 0.2442 |
| CalfRight | 0.2458 | 0.0738 | 0.0437 | 0.1525 | 0.2835 | 0.0426 | 0.03 | 0.1927 |
| FootRight | 0.1237 | 0.0685 | 0.077 | 0.059 | 0.1215 | 0.1012 | 0.0912 | 0.0627 |
| ToeRight | 0.0786 | 0.0257 | 0.0154 | 0.0649 | 0.0595 | 0.0431 | 0.0286 | 0.0483 |
| ToeRightEnd | 0 | 0 | 0 | 0 | 0.001 | 0.0002 | 0.0009 | 0.0008 |

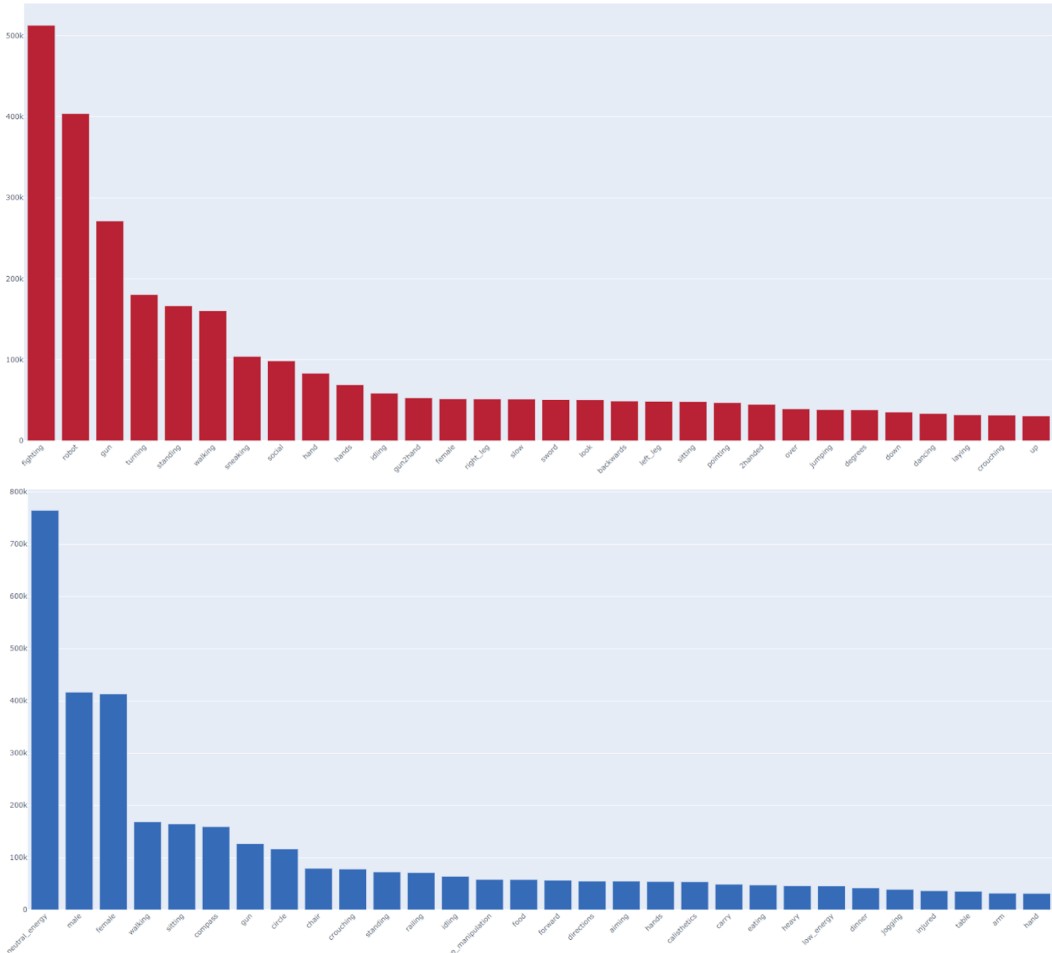

Figure 5: The distribution of tags across frames in the original Mixamo (top) and Anonymous (bottom) datasets.

## E  TRAINING AND EVALUATION SETUP: DETAILS

We use Algorithm 1 of Appendix C to sample batches of size 2048 from the training subset. The number of effectors is sampled once per batch and is fixed for all batch items to maximize data throughput. The training loop is implemented in PyTorch (Paszke et al., 2019) using Adam optimizer (Kingma & Ba, 2015) with a learning rate of 0.0002. Hyperparameter values are adjusted on the validation set (see Appendix E for hyperparameter settings).

We report $\mathcal{L}_{gpd-L2}^{det}$, $\mathcal{L}_{ikd-L2}^{det}$, $\mathcal{L}_{loc-geo}^{det}$ metrics calculated on the test set, using models trained on the training set. $\mathcal{L}_{gpd-L2}^{det}$ is computed only on the root joint. These metrics characterise both the 3D position accuracy ($\mathcal{L}_{gpd-L2}^{det}$, $\mathcal{L}_{ikd-L2}^{det}$) and the bone rotation accuracy ($\mathcal{L}_{loc-geo}^{det}$). They are defined as follows:

$$\mathcal{L}_{gpd-L2}^{det} = \text{MSE}(\mathbf{g}_0, \widetilde{\mathbf{f}}_{R,0}) \tag{22}$$

$$\mathcal{L}_{ikd-L2}^{det} = \sum_{j=1}^{J} \text{MSE}(\mathbf{g}_j, \widehat{\mathbf{g}}_j) \tag{23}$$

$$\mathcal{L}_{loc-geo}^{det} = \sum_{j=1}^{J} \text{GEO}(\mathbf{R}_j, \widehat{\mathbf{R}}_j) \tag{24}$$

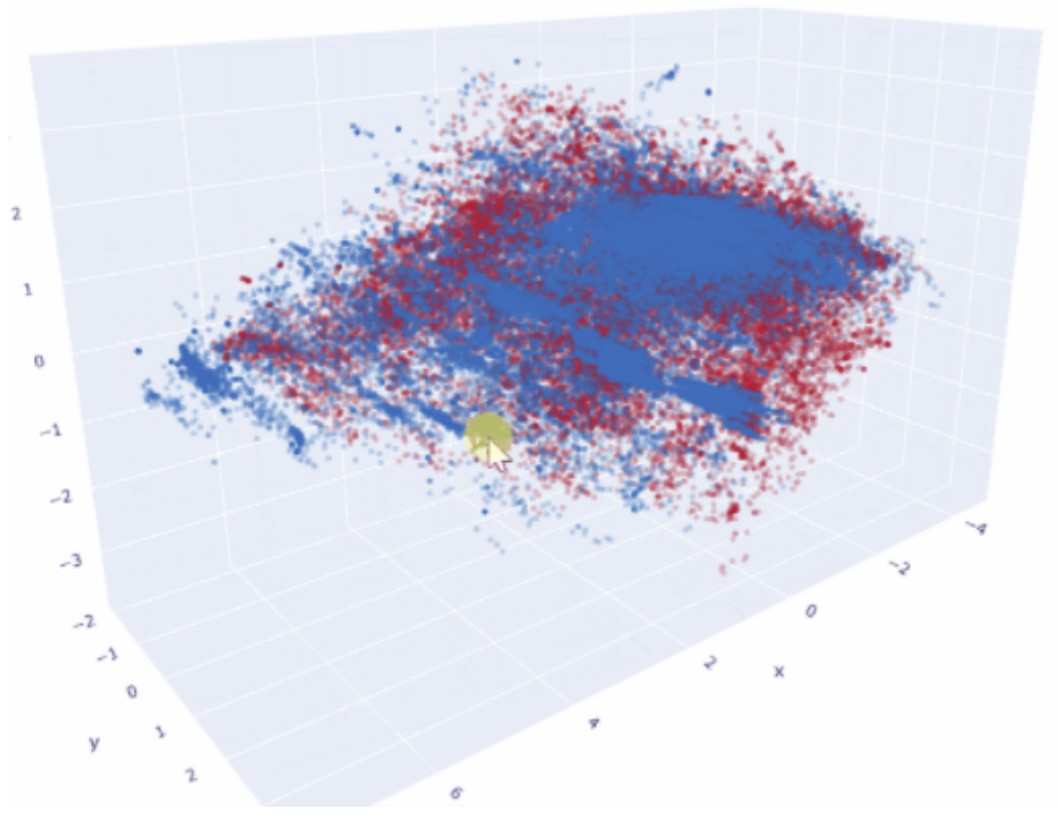

Figure 6: The distribution of PCA of flattened rotations of all joints of miniMixamo (red) and miniUnity (blue) datasets. Each point corresponds to a pose. It is clear that there are both overlapping areas corresponding to clusters with similar poses and areas with disjoint clusters, showing distinct pose configurations characteristic of the two different datasets.

| Hyperparameter | Value | Grid |
|---|---|---|
| Epochs, miniMixamo/ miniUnity | 40k/15k | [20k, 40k, 80k] / [10k, 15k, 40k] |
| Losses | MSE, GEO, LAT | MSE, GEO, LAT |
| Width ($d_h$) | 1024 | [256, 512, 1024, 2048] |
| Blocks ($R$) | 3 | [1, 2, 3] |
| Layers ($L$) | 3 | [2, 3, 4] |
| Batch size | 2048 | [512, 1024, 2048, 4096] |
| Optimizer | Adam | [Adam, SGD] |
| Learning rate | 2e-4 | [1e-4, 2e-4, 5e-4, 1e-3] |
| Base L2 loss scale ($W_{pos}$) | 1e2 | [1, 10, 1e2, 1e3, 1e4] |
| Max noise scale ($\sigma_{M,0}, \sigma_{M,1}$) | 0.1 | [0.01, 0.1, 1] |
| Max effector weight ($W_M$) | 1e3 | [10, 1e2, 1e3, 1e4] |
| Noise exponent, $\eta$ | 13 | 13 |
| Dropout | 0.01 | [0.0, 0.01, 0.05, 0.1, 0.2] |
| Embedding dimensionality | 32 | [16, 32, 64, 128] |
| Augmentattion | mirror, rotation | [mirror, rotation, translation] |

Table 5: Settings of ProtoRes hyperparameters and the hyperparameter search grid.

Here $\widetilde{\mathbf{f}}_{R,0}$ and $\mathbf{g}_0$ are ground truth global location of the root joint and its prediction from the GPD, respectively; $\mathbf{g}_j$ and $\widehat{\mathbf{g}}_j$ ground truth location of joint $j$ obtained by subjecting the rotation prediction

| High Energy | Low Energy | Injured | Neutral Energy | | |
|---|---|---|---|---|---|
| Generic | Generic | Generic | Feminine | Macho | Generic |
| *Female & Male* | *Female & Male* | *Female & Male* | *Female* | *Male* | *Female & Male* |
| Locomotion | Locomotion | Locomotion | Locomotion | Locomotion | Locomotion |
| - Crouch | - Crouch | - Crouch | Crouch | Crouch | - Crawls |
| - Jogs | - Jogs | - Jogs | - Jogs | - Jogs | - Crouch |
| - Runs | - Runs | - Runs | - Runs | - Runs | - Jogs |
| - Walk | - Walk | - Walk | - Walk | - Walk | - Rifle Aim Walk |
| | | | | | - Rifle Crawl |
| | | | | | - Rifle Crouch |
| | | | | | - Rifle Jogs |
| | | | | | - Rifle Walks |
| | | | | | - Runs |
| | | | | | - Walks |
| | | | | | - Idles |
| | | | | | - Rifle Idles |
| | | | | | - Walk Heavy Backpack |
| | | | | | - Walk Heavy Sack |
| | | | | | - Walk Ladder |
| | | | | | - Walk Dragging Object |
| | | | | | Archery |
| | | | | | Bokken Fighting |
| | | | | | Calisthetics |
| | | | | | Door Interactions |
| | | | | | Fist Fighting |
| | | | | | Food |
| | | | | | Handgun |
| | | | | | - Downwards |
| | | | | | - Level |
| | | | | | - Misc |
| | | | | | - Upwards |
| | | | | | - Verticals |
| | | | | | Hands |
| | | | | | Knife Fighting |
| | | | | | Longsword Fighting |
| | | | | | Misc |
| | | | | | Phone |
| | | | | | Place |
| | | | | | Railing Interactions |
| | | | | | Rifle |
| | | | | | Seated Interactions |
| | | | | | Shotgun |
| | | | | | Standing |
| | | | | | Sword |
| | | | | | Wall |

Table 6: The hierarchy of motion styles and categories in miniUnity. *Female & Male* indicate that every clip of every underlying category has been captured with both a female and a male actor.

from IKD to the forward kinematics process; $\mathbf{R}_j$ and $\widehat{\mathbf{R}}_j$ are ground truth local rotation matrix of joint $j$ and its prediction obtained from the IKD, respectively.

The evaluation framework tests model performance on a pre-generated set of seven files containing 6, 7, ..., 12 effectors respectively. Skeleton is split in six zones, with four main zones including each limb, the hip zone and the head zone. In each file, we first sample one positional effector from each main zone. Remaining effectors are sampled randomly from all zones and effector types, mimicking pose authoring scenarios observed in practice. Metrics are averaged over all samples in all files, assessing the overall quality of pose reconstruction in scenario with sparse and variable inputs. All tables present results averaged over 4 random seed retries and metric values computed every 10 epochs over last 1000 epochs, rounded to the last statistically significant digit.

**Hyperparameter settings** The training loop is implemented in PyTorch (Paszke et al., 2019) using Adam optimizer (Kingma & Ba, 2015) with a learning rate of 0.0002. We tried to use SGD optimizer

| Motion Category |
| --- |
| Combat |
| Adventure |
| Sport |
| Dance |
| Fantasy |
| Superhero |

Table 7: Motion categories in miniMixamo

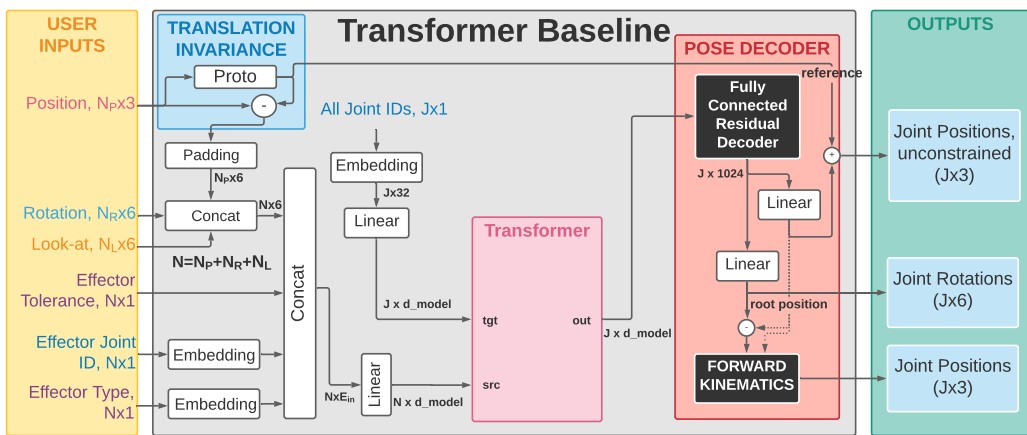

Figure 7: Block diagram of the Transformer baseline architecture.

to train the architecture, but it was very difficult to obtain stable results with it. Adam optimizer turned out to be much more suitable for our problem. The learning rate was selected to be 0.0002, which is lower than Adam's default. Obtaining stable training results with higher learning rates was not feasible. Batch size is selected to be 2048 to accelerate training speed. In practice we observed slightly better generalization results with smaller batch size (1024 and 512). The detailed settings of ProtoRes hyperparameters are presented in Table 5.

## F  MASKED-FCR BASELINE ARCHITECTURE

Masked-FCR is a brute-force unstructured baseline that uses a very wide $J \cdot 3 \cdot 7$ input layer ($J$ joints, 3 effector types, 6D effector plus one tolerance value) to handle all possible effector permutations. Each missing effector is masked with one of $3 \cdot J$ learnable 7D placeholders. Masked-FCR has 3 encoder and 6 decoder blocks to match ProtoRes.

## G  TRANSFORMER BASELINE ARCHITECTURE

We implement Transformer baseline using the default Transformer module available from PyTorch `https://pytorch.org/docs/stable/generated/torch.nn.Transformer.html`. The block diagram of the Transformer baseline architecture is shown in Fig. 7.

We use a standard transformer application scenario in which the transformer source input is fed with the variable length input and the required outputs are queried via target input. In our case the variable length input corresponds to the effector data concatenated with embedded effector categorical variables. The query for the output consists of the embeddings of all joints. Note that the joint embedding is reused both for source and target inputs and both inputs are projected to the internal d_model dimensionality of transformer.

| Hyperparameter | Value | Grid |
|---|---|---|
| FCR decoder parameteres | | |
| FCR Blocks ($R$) | 6 | N/A |
| FCR Width ($d_h$) | 1024 | N/A |
| Transformer parameteres | | |
| d_model | 128 | [64, 128, 256] |
| nhead | 8 | [1, 2, 4, 8] |
| num_encoder_layers | 2 | [1,2,3] |
| num_decoder_layers | 2 | [1,2,3] |
| dim_feedforward | 1024 | [256, 512, 1024] |
| dropout | 0.01 | 0.01 |
| activation | relu | relu |
| Base L2 loss scale ($W_{pos}$) | 100 | [10, 100] |

Table 8: Settings of Transformer baseline hyperparameters and the hyperparameter search grid.

The embeddings of joint IDs are used to query the Transformer output, producing one encoder embedding for each of $J$ joints. The $J$ encodings are fed into the 6-block FCR decoder (to match the total number of decoder blocks in ProtoRes) with two heads: one predicting rotation and one predicting unconstrained position. This is similar to the use of Transformer to predict bounding box class IDs and sizes for object detection (Carion et al., 2020). Predictions of rotations and of the root joint are used in the forward kinematics pass, just as in ProtoRes.

Internally, Transformer processes both source and target inputs via self-attention first and then applies the multi-head attention between source and target after self-attention. This results in the output embedding for each skeleton joint that depends on all the input information as well as the learned interactions across all output skeleton joints. The output embedding of the transformer is then decoded to unconstrained position and rotation outputs using a two-headed Fully-Connected residual stack (this is the same architecture as the one used in ProtoRes decoders). Note that this is a well-known Transformer application scheme that has recently been used to achieve SOTA results in object detection, for example (Carion et al., 2020). Table 8 lists the hyperparameter settings for the Transformer baseline. Note that only hyperparameters that are unique to this baseline or different from the ProtoRes defaults appearing in Table 5 are listed.

## H ABLATION OF THE DECODER: DETAILS

The detailed results of the decoder ablation are shown in Table 9. One block of decoder is much less computationally expensive than one block of encoder. One block of encoder processes $N$ effectors, whereas the decoder deals with the partially defined pose representation collapsed to a vector. Hence the decoder block is $N$ times less expensive. To demonstrate the effectiveness of decoder, we keep all hyperparameters at defaults described in Section 3.2 and vary the number of encoder and decoder blocks in ProtoRes (0 decoder blocks corresponds to a simple linear projection of encoder output). We measure the train time of each configuration on NVIDIA M40 24GB GPU installed on Dell PowerEdge R720 server with two Intel Xeon E5-2667 2.90GHz CPU. The table reveals a few things. First, adding more decoder blocks significantly increases accuracy when the number of encoder blocks is lower (e.g. 3 or 5). When the number of encoder blocks is high (e.g. 7) linear projection provides similar accuracy. Second, using non-trivial decoder is computationally more efficient. For example, the 3 encoder and 3 decoder blocks configuration has comparable accuracy with 5 encoder and 1 decoder blocks, however it is noticeably more compute efficient (5+1 configuration uses 40% more compute time than 3+3).

Table 9: Ablation of the decoder. Random benchmark, lower values are better. Train time is measured on a 2GPU Dell PowerEdge R720 server with two NVIDIA M40 24GB GPUs and two Intel Xeon E5-2667 2.90GHz processors, each GPU running one ProtoRes training session.

| | | miniMixamo | | | | miniUnity | | | |
|---|---|---|---|---|---|---|---|---|---|
| Encoder blocks | Decoder blocks | $\mathcal{L}_{gpd-L2}^{det}$ | $\mathcal{L}_{ikd-L2}^{det}$ | $\mathcal{L}_{loc-geo}^{det}$ | Train time, h | $\mathcal{L}_{gpd-L2}^{det}$ | $\mathcal{L}_{ikd-L2}^{det}$ | $\mathcal{L}_{loc-geo}^{det}$ | Train time, h |
| 3 | 0 | 1.54e-3 | 4.59e-3 | 0.2485 | 91 | 1.05e-3 | 3.65e-3 | 0.1939 | 105 |
| 3 | 1 | 1.35e-3 | 4.34e-3 | 0.2433 | 95 | 0.93e-3 | 3.52e-3 | 0.1895 | 110 |
| 3 | 2 | 1.34e-3 | 4.24e-3 | 0.2397 | 102 | 0.93e-3 | 3.34e-3 | 0.1840 | 116 |
| 3 | 3 | 1.36e-3 | 4.16e-3 | 0.2381 | 106 | 0.93e-3 | 3.28e-3 | 0.1817 | 121 |
| 5 | 0 | 1.27e-3 | 4.20e-3 | 0.2399 | 144 | 0.84e-3 | 3.27e-3 | 0.1824 | 166 |
| 5 | 1 | 1.28e-3 | 4.30e-3 | 0.2390 | 148 | 0.81e-3 | 3.24e-3 | 0.1818 | 171 |
| 5 | 2 | 1.15e-3 | 4.02e-3 | 0.2345 | 153 | 0.77e-3 | 3.10e-3 | 0.1791 | 176 |
| 5 | 3 | 1.18e-3 | 4.03e-3 | 0.2351 | 157 | 0.82e-3 | 3.07e-3 | 0.1785 | 182 |
| 7 | 0 | 1.13e-3 | 4.00e-3 | 0.2355 | 196 | 0.74e-3 | 2.98e-3 | 0.1762 | 226 |
| 7 | 1 | 1.23e-3 | 4.16e-3 | 0.2356 | 200 | 0.79e-3 | 3.07e-3 | 0.1780 | 230 |
| 7 | 2 | 1.13e-3 | 4.51e-3 | 0.2383 | 205 | 0.78e-3 | 3.10e-3 | 0.1780 | 236 |
| 7 | 3 | 1.15e-3 | 3.98e-3 | 0.2352 | 209 | 0.82e-3 | 3.14e-3 | 0.1783 | 241 |

## I ABLATION OF THE PROTOTYPE-SUBTRACT-ACCUMULATE STACKING PRINCIPLE

Here we compare the proposed Prototype-Subtract-Accumulate (PSA) residual stacking scheme described in equations (2)-(5) against the ResPointNet (Niemeyer et al., 2019) stacking scheme: Maxpool-Concat Daisy Chain (MCDC). To implement the MCDC stacking proposed by Niemeyer et al. (2019) we setup the experiment as follows.

- Equation (2) is replaced with the concatenation of the output of the previous block, $\mathbf{b}_r$, with the maxpool of the previous block output along axis 1 (we use batch, effector, channels convention for tensor axes 0,1,2; respectively)

- In equation (4) we only compute $\mathbf{b}_r$, since $\mathbf{f}_r$ is not used

- Equation (5) is removed

- The final pose embedding is created using the maxpool along axis 1 of $\mathbf{b}_r$ at the last encoder block

- We use decoder with 0 blocks, *i.e.* only the linear projection at the end to derive both PSA and MCDC results. This is because (i) residual decoder is not part of the original design by Niemeyer et al. (2019), (ii) in this study we focus exclusively on the effects of stacking within the encoder, equalizing all other experimental conditions.

- Hyperparameters of both architectures are taken from Table 5 in Appendix C

We show that our stacking scheme is more accurate and allows stacking deeper networks more effectively. Quantitative results are shown in Table 10. It is clear that the proposed stacking approach provides gain in setups with varying number of encoder blocks. For example, in the case of small number of 3 blocks (computationally efficient setup) our PSA approach is clearly more accurate than MCDC approach of (Niemeyer et al., 2019). Similarly, with 7 blocks (training time is more than two times longer) PSA is again more accurate than MCDC. Moreover, in the case of PSA we see noticeable accuracy improvement while increasing the number of encoder blocks from 5 to 7, whereas MCDC provides almost no additional gain beyond 5 blocks. We conclude that the proposed PSA stacking mechanism is better than MCDC as it provides better accuracy in the computationally efficient configuration and it is significantly more effective at supporting deeper architectures that provide better accuracy in the case of PSA, whereas MCDC accuracy saturates at 5 blocks providing little to no accuracy gain with deeper architectures.

Table 10: Ablation of the Prototype-Subtract-Accumulate. Random benchmark, lower values are better.

| Stacking scheme | Encoder blocks | miniMixamo | | | miniUnity | | |
| --- | --- | --- | --- | --- | --- | --- | --- |
| | | $\mathcal{L}_{gpd-L2}^{det}$ | $\mathcal{L}_{ikd-L2}^{det}$ | $\mathcal{L}_{loc-geo}^{det}$ | $\mathcal{L}_{gpd-L2}^{det}$ | $\mathcal{L}_{ikd-L2}^{det}$ | $\mathcal{L}_{loc-geo}^{det}$ |
| MCDC | 3 | 1.52e-3 | 4.65e-3 | 0.2539 | 1.12e-3 | 3.93e-3 | 0.1984 |
| MCDC | 5 | 1.28e-3 | 4.25e-3 | 0.2395 | 0.85e-3 | 3.25e-3 | 0.1831 |
| MCDC | 7 | 1.28e-3 | 4.24e-3 | 0.2384 | 0.85e-3 | 3.26e-3 | 0.1830 |
| PSA (**ours**) | 3 | 1.54e-3 | 4.59e-3 | 0.2485 | 1.05e-3 | 3.65e-3 | 0.1939 |
| PSA (**ours**) | 5 | 1.27e-3 | 4.20e-3 | 0.2399 | 0.84e-3 | 3.27e-3 | 0.1824 |
| PSA (**ours**) | 7 | 1.13e-3 | 4.00e-3 | 0.2355 | 0.74e-3 | 2.98e-3 | 0.1762 |

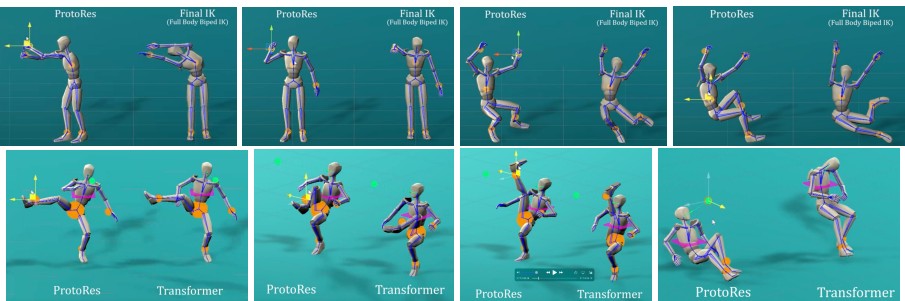

Figure 8: Qualitative comparison results. ProtoRes vs. FinalIK (top row). ProtoRes vs. Transformer (bottom row).

## J  QUALITATIVE COMPARISON TO TRANSFORMER AND FINALIK BASELINES

In this section we provide additional qualitative comparison between ProtoRes and two baselines, the non-learnable baseline FinalIK and the machine learning baseline Transformer. The FinalIK comparison results appear in the top row of Figure 8. The Transformer comparison results appear in the bottom row of Figure 8.

The characteristic feature of FinalIK is that it lacks inductive bias towards realistic poses. Therefore it is very easy to create effector configurations that result in unnatural poses as can be seen in Figure 8. Conversely, creating realistic poses requires significant amount of tuning of the individual joints as most joints of the body act very independently and locally. With FinalIK, it is relatively hard to produce a believable pose with only a few effectors. ProtoRes fills this gap by providing a learned prior for realistic poses. As a result, most of the poses created by ProtoRes look natural, no matter how many effectors are used to steer the pose. In addition, the effector space can be augmented dynamically to capture missing accents, if necessary.

When comparing ProtoRes against Transformer, we can see that Transformer has a less global approach to forming a pose than ProtoRes. This manifests itself in the Tranformer based model having a tendency to create poses in which limbs penetrate each other or the rest of the body (see Figure 8 bottom row, pictures 1,2,3 from the left). Also, Transformer is very good at following the effector inputs and reproducing them in the reconstructed pose — so much that it is willing to sacrifice the overall final pose plausibility at the expense of sticking to the effector, taking the effector guidance very "literally" and missing its re-interpretation given global context. A good example of this behaviour is presented in Figure 8 bottom row, rightmost picture. We see that ProtoRes brings the entire body to the floor and prepares the arms to touch the floor to support the body, all in the attempt to produce a plausible pause of someone looking at the look-at target that is placed relatively low. On the other hand, Transformer takes the look-at effector guidance very literally and locally, willing to break the neck of the pose to look at the provided target and not realizing that the entire body position needs to be changed to accommodate for this rather peculiar configuration of effectors to create a believable pose. In general, our observation is that the Transformer based baseline response to effector inputs tends to be localized. As a consequence, its behaviour has a very pronounced pure

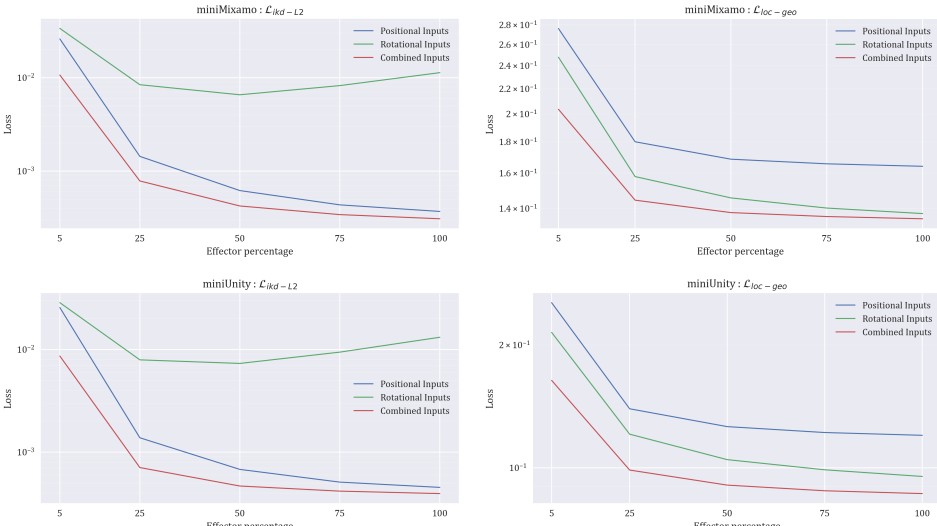

Figure 9: Metrics as a function of the number and type of effectors. miniMixamo dataset position and rotation metrics (top), miniUnity dataset position and rotation metrics (bottom). Position only effectors (blue), rotation only effectors (green), rotation and position effectors (red). Position error metric (left) and rotation error metric (right). The *x*-axis shows the percentage of joints used as effectors, the *y*-axis shows the tracked metric value.

IK flavour to it, at times reminding the behaviour of FinalIK, especially when facing difficult effector configurations. A lot of the time, however, the outputs of ProtoRes and Transformer are comparable and consistent, since both of them develop strong inductive bias towards realistic poses.

## K    THE EFFECTS OF THE NUMBER AND TYPE OF EFFECTORS

The results of the experiment studying the effects of the number of input effectors on the reconstruction accuracy are shown in Figure 9. In this experiment, we sample the inputs to the neural network uniformly at random as in the main text, but now we specifically control the type and the number of effectors that are being provided at the input of the network and we study the evolution of error metrics as a function of the number of provided effectors. A few interesting observations emerge in this experiment. We see that in most cases, errors decrease monotonically as the percentage of joints used as effectors increases, however, for most of them we also observe saturation and the errors do not converge to zero. For the position error and the case of rotation only inputs, we see a U-shape behaviour. Let us reflect on these observations in order.

The case when both position and rotation inputs are provided, forcing error to go down, is intuitively clear, because more effectors provide more information to the input and thus more accurate pose reconstructions are achieved. The saturation of the error may happen for a few plausible reasons, including: (i) the network is trained with effectors whose number is sampled uniformly at random in the range $[5, 16]$ (5-25% of joints are used as effectors), clearly above 25% the network is operating in the out-of-distribution regime potentially leading to sub-optimal results, (ii) the network operates as a noisy auto-encoder and it is specifically trained to reconstruct a good, but potentially imperfect pose, therefore we can never expect it to reach perfect reconstruction, even if all joints are provided as inputs. Error saturation is not something we expect to happen for traditional IK systems such as FinalIK, which when provided with enough effectors for a valid pose should be able to recover all parameters of the pose through mathematical equations, resulting in zero errors. At the same time, it is worthwhile noting that (i) the operation with full pose input is a use case neither for FinalIK nor for ProtoRes, in fact the typical use case is to reconstruct full pose from the least number of inputs and (ii) if smaller error is desired with full pose inputs, the network can be trained to address this specific scenario.

The case when only rotation inputs are provided (green curve) is interesting in that we see the rotation errors going down with increasing number of effectors (right pane) and the position errors following

a U-shape (left pane). It appears that the network is able to use more information about rotations to reduce rotation errors, but the task of reconstructing positions from only rotational inputs is too hard. This is not surprising, as rotations of joints bear little information about the absolute position of the skeleton in space, which is necessary to reconstruct the global position of the root joint. To aggravate the situation, errors in the global root joint position will be magnified during the forward kinematics pass used to compute global positions of all joints. It looks like when the network is provided with a relatively small number of rotational inputs (5-25%) it is still able to focus on reconstructing global root joint position, whereas providing more rotational inputs makes errors worse (U-shape on the green line in the left pane). We conjecture that two adverse factors can be at play here: (i) increasingly out-of-distribution operation makes processing large number of rotational inputs less effective and (ii) the network is trained to treat provided effectors as high-priority for reconstruction, therefore it will concentrate on minimizing rotation errors while de-prioritizing positions, eventually rotation errors will decrease at the expense of growing position errors, which is clear comparing green curve in the left and in the right panes. In fact, a similar effect can be observed on the position only effectors curve (blue), with the exception that there is no U-shape on the rotation losses, but there is definitely a lot more saturation in the blue curve on the right (rotation error of position only inputs) than in the blue curve on the left (position error of position only inputs). It looks like position inputs tend to provide more information about rotations to be reconstructed than the rotation inputs do about positions to be reconstructed, which makes sense based on the geometry of the problem.

## L    VIDEOS AND DEMONSTRATIONS

We provide here descriptions related to each video found in the supplementary materials. The Unity tool that is used to produce demo videos and qualitative pictures shows the raw neural network outputs without applying any post-processing. This is done to ensure fair comparison of different methods and avoid any misleading results due to post-processing. Note that most of the demonstrations are done using a ProtoRes model trained on a large internal dataset not evaluated in this work. One of our demonstrations, described below in Appendix L.5 qualitatively shows some of the most severe impacts of training on ablated datasets.

### L.1    PROTORES DEMO

The video presents an overview of the integration of ProtoRes as a posing tool inside the Unity game engine, showcasing how different effector types can be manipulated and how they can influence the resulting pose. In this demo, a user-defined configuration is presented in the UI, allowing one to choose which effectors are enabled within that configuration. Note that this configuration could contain more or less effectors of each type, and can be built for specific posing needs. The most generic configuration would present all possible effectors inside each effector type sub-menu.

### L.2    POSING FROM IMAGES

This video presents screen recordings of a novice user using ProtoRes to quickly prototype poses taken from 2D silhouette images. Note that one can reach satisfactory results in less than a minute in each case, with a relatively low number of manipulations. Note also that fine-tuning the resulting poses can always be achieved by adding more effectors and applying more manipulations.

### L.3    LOSS ABLATION

This recording shows a setup where different models are used with identical effector setup. Both models use the ProtoRes architecture. On the left, the model uses the total loss presented in Algorithm 1, whereas the right-hand side model uses positional losses only (GPD and IKD positional losses) and both local and global rotational losses, as well as look-at losses are disabled. This demonstration clearly shows how positional constraints, even when respected, do not suffice to produce realistic human poses. Joint rotations have to be modeled as well.

### L.4 FINALIK COMPARISON

This recording shows a setup where ProtoRes is compared to a full body biped IK system provided by FinalIK RootMotion (2020), with an identical effector setup. Note that FinalIK solves constraints by modifying the current pose, often resulting in smaller changes in the output, when compared to ProtoRes that predicts a full pose at each update. The lack of a learned model of human poses in FinalIK becomes quickly noticeable when manipulating effectors significantly.

### L.5 DATASETS COMPARISON

In these demonstrations, we showcase how training ProtoRes on different datasets can impact the results. We showcase models trained on the two ablated datasets presented in this work, i.e. miniAnonymous (left) and miniMixamo (right). We also show performance of a model trained on the full Mixamo Adobe Inc. (2020) dataset (center) to qualitatively show how performance can be improved with more training data. In all of these recordings, one can notice differences in the resulting poses, emphasizing the fact that human posing from few effectors, when no extra conditioning signals are used, is an ambiguous task that will be influenced by the training data.

The first sequence makes this fact especially obvious on the finger joints and the head's look-at direction. The second sequence shows how good data coverage in the training set can significantly impact performance in special or rare effector configurations. Finally, the third sequence shows a similar pattern for look-at targets, where the difference in training data can be noticed in the general posture of the character and the varying levels of robustness with respect to those targets.

### L.6 RETARGETING

This video shows that ProtoRes trained on one dataset can be successfully applied to multiple characters defined on skeletons on which our model was never trained. Currently, the model is trained on a dataset with a specific skeleton layout with specific bone offsets. In a naïve application scenario, a new skeleton would require retraining the model on a dataset for this specific skeleton, which could be a prohibitively expensive exercise. To overcome this limitation, we show in this video that the model learned using our approach can be retargeted to very different skeletons using conventional retargeting methods via a process that is fully automatic, without changing the model in any way. Thus, our model can be seamlessly used without any retraining with a wide variety of skeletons. In practice this solution allows the use of the same model on characters having similar skeleton in terms of topology and/or morphology and very different in other aspects such as proportions (bone lengths) or bone count. For instance, as our demo shows, it can be used with any humanoid character and still produces valuable results. Transfer of the model across skeletons using more sophisticated ML-based techniques such as fine-tuning, conditioning, domain adaptation is deemed to a be a fruitful future research area, for which our current results form a strong baseline.

### L.7 APPLICATION TO THE QUADRUPED SKELETON

In this video we demonstrate that ProtoRes can be applied to a completely different type of skeleton (quadruped) and it does not require any additional coding as opposed to FinalIK (no hand-crafted skeleton, effectors, bone chains, pulling, etc). It does require data and re-training however, but does not contain anything specific to humanoids. In fact, we have successfully trained the proposed model on a quadruped (dog) dataset without changing the model or hyperparameters, which demonstrates the generality of the method.

### L.8 COMPARISON TO TRANSFORMER

In this video we demonstrate qualitative examples demonstrating the basic difference between Transformer and ProtoRes. First, generally speaking, in many cases with challenging effector settings ProtoRes shows much greater robustness to outliers with respect to pose plausibility, whereas in many cases Transformer's behaviour feels like that of a pure non-learnable IK model. Second, Transformer has a tendency to generate self-intersections between the limbs to satisfy some of the constraints more closely at the expense of the overall pose plausibility. Finally, in response to an extreme look-at constraint Transformer can bend the neck behind the body, which in reality never happens in the data

and would break the neck of a real human. In contrast, ProtoRes keeps the neck naturally oriented when the look-at target is unattainable.

From the theoretical perspective, the fundamental difference between the Transformer framework and the ProtoRes is as follows. ProtoRes creates one global representation of the partially specified pose and then reconstructs the full pose in one shot through a global IKD. On the other hand, Transformer generates embeddings of individual joints for the full output pose, from which a shared IKD reconstructs the local rotation angles for each joint. Transformer has all the ingredients to reproduce processing similar to our approach. However, in reality the Transformer takes a different learning route and (i) learns more localized joint predictions and (ii) learns to more strictly respect local input constraints, even at the expense of creating poses that are not statistically plausible. Combining the global (ProtoRes) and local (Transformer) approaches to derive better hybrid models seems like a promising direction for future work.

## L.9 LIMITATIONS

The final video shows examples of some specific limitations of the approach that are listed in Appendix M. Namely, we first expose specific consequences of the lack of temporal consistency in our problem formulation, where smoothly moving an effector can cause flickering on some joints, such as the fingers. We also show how between some effector configurations, the character must be flipped completely to stay in a plausible pose, and how it's possible to place some effectors to reach an invalid pose coming from that *flip region* of the latent manifold. Finally, we showcase some problematic behaviors that can be caused by extreme look-at targets. In some cases, especially with many other constraints, ProtoRes will tend to produce a plausible pose that will not respect the look-at constraint. In other cases, the extreme look-at target may cause an unrealistic pose, e.g. by causing the character to have an impossible neck rotation. It is interesting to note how invalid poses from look-at effectors tend to happen more often than from other effector types with novice users. We hypothesize that the plausible region of a look-at target, given a current character pose, is less intuitive to grasp than for other effector types. Indeed, the current pose of the character seems to guide more precisely the placement of positional and rotational effectors than look-at effectors, leading more often to configurations outside of the training distribution for look-at effectors.

## M LIMITATIONS

The limitations of our work can be summarized as follows:

- Constraints are not satisfied exactly, as opposed to the conventional systems. The limitation can be observed in the demo video 4_Final_IK_comparison.mp4, which compares ProtoRes and FinalIK side-by-side. For example, at seconds 18-21 we can see the chest position is not satisfied exactly by ProtoRes and the generated left foot position traverses the vicinity of the constraint as the user changes the position of the left foot. Another example like this can be seen at seconds 35-41, in which the left foot effector is not satisfied exactly by ProtoRes. It is interesting that for the rather peculiar configuration of effectors, FinalIK produces outputs that tend to severely twist the joints, perhaps beyond the capabilities of an average human. At the same time ProtoRes tends to trade the precision of following individual effectors with the plausibility of the overall pose. This is the price to pay for the ability of the model to inject the data-driven inductive bias that can be used to reconstruct pose from very sparse inputs. This could be mitigated using a conventional solver on top of the trained model. In this case, the model will produce a globally plausible pose, whereas the solver will only do the final pass to strictly satisfy certain constraints. Also, to provide additional flexibility in solving some of the constraints more strictly than the others, our model provides an effector tolerance mechanism that can help the user trade off the strictness of satisfying certain effectors vs. some others.

- Lack of temporal consistency. Our work solves the problem of creating a discrete pose. Therefore, it is limited in how it can be applied to modify an underlying smooth animation clip. For example, we can see flickering of joints (especially fingers) when effectors follow an underlying smooth animation (the finger embedding space is not smooth and has a high

ambiguity). This happens to a smaller degree with the head when it is not constrained with look-at or rotation inputs.

- Exotic poses significantly deviating from the the training data distribution (a common ML/DL problem) may be hard to achieve. For example, the Lotus yoga pose is very hard to achieve with small number of effectors. Extreme or rare effector configurations may not be respected. Extreme look-at targets may not be followed or can cause artifacts in the resulting pose.

- Some effector displacement can cause a complete flip in the final pose as it makes more sense to be e.g. left-oriented or right-oriented to reach a hand position. This is normal, but we can sometimes reach "in-between" poses on the boundary of the hand effector that causes the flip, leading to weird poses

- We also noted a limitation as "aiming" poses (holding something in the hands). For example, Finger poses are generally wrong w.r.t. to a gun without additional finger constraints. It may be cumbersome to place hands + look-at for each aiming pose/angle?

- Runtime. In its current state, the model allows interactive real-time rate (about 100 FPS). This is very good for the primary application area of the model in the interactive pose design. However, the current model cannot be used for runtime applications such as driving charachters directly in real-time games, because it would consume too high of a time budget (about 10ms, which is too much to be usable in the game runtime context).

- No contextual input is supported (text description or environment awareness), in particular for finger posing and feet collisions

- Good for realism, but might limit creativity. In particular, no bone stretching support, which is sometimes used by animators to add more expressiveness to non-realistic characters.

- The current model struggles when a large number of fine-grain controls, especially fingers are used simultaneously. Perhaps, a more structured hierarchical approach can be used to enable this functionality.

