# OpenReview forum: "ProtoRes: Proto-Residual Network for Pose Authoring via Learned Inverse Kinematics"
_ICLR.cc/2022/Conference — ICLR 2022 Oral_

### Official Review · Reviewer_5Kos · 2021-10-31

**Correctness:** 3
**Technical Novelty And Significance:** 3
**Empirical Novelty And Significance:** 3
**Recommendation:** 6
**Confidence:** 3

**Main Review:**

Pros:
1. The paper solves an interesting and long-standing inverse-kinematics problem. To solve the problem, the paper novelly proposes a neural-network-based method to automatically handle the problem.
2. In general, the paper is technically sound with experiments to support its argument. Additionally, the paper proposes two datasets to train and validate the proposed methods.
2. In general, the paper is easy to follow.

Cons:
1. The organization of the paper could be further improved. For example, the definition of 6D rotations is only explained in Sec 1.2. It would be better to mention it the first time this term appeared, otherwise, it is slightly confusing. Besides, it is suggested to merge the five contributions into three or four. e.g. merge the fourth and fifth contributions.
2. Some important details are missing. a). How are the metrics L^{det}_{gpd}, L^{det}_{ikd}, L^{det}_{loc} defined and what are the units of them? b). How are the two datasets split into train and evaluation?
3. It would be great if the paper could show more qualitative results. For example, the qualitative comparison between the proposed method and the previous methods.
4. It would be great if the paper could show more analysis on the relation between inputs and model performances. For example, how is the performance influenced by the number of input joints and input types (joint positions or joint rotations which are better)?
5. I have a concern about the overall framework. The paper resolves an ill-posed problem, i.e. there could be multiple feasible outputs given the one input. However, the paper uses the encoder-decoder framework which is only able to generate one output given the input. Therefore, I am wondering would it be better to change the current AE architecture to VAE so that multiple outputs could be produced? I would like to hear about the authors' options.

**Summary Of The Paper:**

This paper proposes a DNN-based framework to handle the challenging task of recovering 3D human poses (joint positions and local rotations) from sparse and variable user inputs (joint positions, orientations of a subset of whole body joints). The paper proposes a neural-network architecture with an encoder to extract pose encodings from partial user inputs and a decoder to generate complete poses from the encoded features. The paper further proposes two new datasets to train and evaluate the proposed models. Experimental results demonstrate the superiority of the proposed framework over previous methods.

**Summary Of The Review:**

This paper is interesting and technically sound in general. Therefore, my current rating is boardline accept.

---

> ### Author Response · Authors · 2021-11-16
> **Author response to Reviewer 5Kos 3/3**
>
> **Comment 5**
> I have a concern about the overall framework. The paper resolves an ill-posed problem, i.e. there could be multiple feasible outputs given the one input. However, the paper uses the encoder-decoder framework which is only able to generate one output given the input. Therefore, I am wondering would it be better to change the current AE architecture to VAE so that multiple outputs could be produced? I would like to hear about the authors' options.
>
> **Response 5**
> We thank the reviewer for pointing this out. Indeed, generative/distributional modeling sounds like a very interesting venue for future research. A few detailed points follow.
>
> - We would like to point out that architecturally, our approach can be modified straightforwardly into a generative model by predicting $\mu$ and $\sigma$ from the pose embedding, incorporating the KL divergence term and decoding pose from the associated latent variable, for example. However, there are a couple of problems that need a significant amount of work to make the generative approach viable and useful in our application scenario.
> - First and foremost, we need to develop effective user interface tools which would allow the user to explore the random outputs that the generative model might be able to produce without overwhelming the user with noise. We need to understand how to best present the noisy results to the user (density maps, a few potential poses, one pose per click etc.) and how to enable the user to navigate/select generated pose samples. This is an exciting and non-trivial problem in itself. Unfortunately, until it is solved, the practical use of generative models for our specific problem is very limited. The focus of our current work is to show that the problem of producing realistic human poses using a small set of heterogeneous variable user inputs has a computationally efficient, elegant and practical solution via learned inverse kinematics and neural modeling approaches. We believe that our results validate this and open up opportunities for future research, including generative modeling of partially defined human poses.
> - Second, the performance of generative models would need to be benchmarked to assess the quality of distributional modeling/predictions and provide an objective benchmark to support future model development in this area. Many existing works focus on demonstrating the prediction accuracy of generative models or evaluating the likelihood values of their predictions. We believe that more work needs to be done in this direction to make the assessment of generative models more thorough and objective. For example, high accuracy can be achieved by approaches that do not rely on generative modeling. Likelihood value assessment is not ideal either: the model that only generates samples with high likelihood (and scores well) basically is bound to produce only the most common values, which may not be attractive from the user perspective. The model that produces too many samples in the tails of distribution might not be good either, because it will be hard to steer by the user and will produce too many samples that do not fit the user's guidance signals. More research needs to be done in this direction to identify suitable ways of quantifying the diversity of samples, the adequacy of the overall distribution fit as well as the usefulness of the randomly generated samples for the user of the system. We believe this is a very exciting area for future research.

---

> ### Author Response · Authors · 2021-11-16
> **Author response to Reviewer 5Kos 2/3**
>
> **Comment 3**
> It would be great if the paper could show more qualitative results. For example, the qualitative comparison between the proposed method and the previous methods.
>
> **Response 3**
> We thank the reviewer for their interest in our qualitative results. To address the comment, in the revised manuscript we included an additional Appendix J presenting a few examples comparing FinalIK and Transformer against ProtoRes and discussing in detail the nuances of posing results of the three approaches. These complement our supplementary videos that contain much richer qualitative comparisons. Unfortunately, this year there is no extra page provided for addressing the comments. The current manuscript has already undergone a few compacting iterations in the attempt to strike the balance between being well-motivated, self-contained from the technical perspective as well as providing some intuitive visuals and supporting claims with strong quantitative evidence. Unfortunately, we have not been able to identify the pieces in the current manuscript that could be removed to provide space for the contents of the new Appendix J. We will be more than happy to look at this issue again if Reviewer 5Kos provides additional opinion on which parts of the main body could be removed to accommodate the material in Appendix J, without compromising the clarity of the manuscript.
>
> **Comment 4**
> It would be great if the paper could show more analysis on the relation between inputs and model performances. For example, how is the performance influenced by the number of input joints and input types (joint positions or joint rotations which are better)?
>
> **Response 4**
> We thank the reviewer for the insightful comment. To address the comment we will include an additional Appendix showing the effect of the number and type of effectors on the loss terms. From our previous preliminary experiments we know for a fact that increasing the number of effectors leads to improved accuracy, which is an intuitively appealing result. In our current implementation we are only logging the aggregate results, so collecting the associated statistics will require additional code modifications and rerunning the training session. So we need some extra time to produce the results. For the effect of the type of effector, we have not conducted experiments to measure the effects. Our randomized evaluation framework uses randomly sampled effector configurations, therefore disentangling the effector effects requires additional regression analysis. Again, this requires extra coding and time to compute the results and we need some extra time to provide them. We will include these results in the final version of the revised manuscript.
>
> Meanwhile, some observations from our practical experience with these models have emerged:
>
>
> - The flexible number of inputs to the model allows for a user to easily adapt the inputs to reach the desired level of precision. In Figure 3 of the manuscript, the four leftmost images show this process, where new effectors are iteratively added to match a desired pose evermore precisely.
>
> - Although we have no precise results yet on the final metrics with respect to input types (e.g. positions vs rotations), a universal conclusion that stems from our own use of the model and from skilled animators trying the tool is that positional effectors are more intuitive to use and accelerate initial pose authoring compared to rotations. Rotations and look-at targets tend to be used to fine-tune a pose authored first from positions. In other words, all types of effectors are useful and act synergistically: positions are often preferred to quickly author a base pose, rotations/lookat are often used in the finetuning phase.

---

> ### Author Response · Authors · 2021-11-16
> **Author response to Reviewer 5Kos 1/3**
>
> We would like to sincerely thank **Reviewer 5Kos** for careful reading of the manuscript and a positive and encouraging review of our work. We address the points raised by the Reviewer in detail below. Please do not hesitate to post additional comments and questions, we will be more than happy to address them.
>
> **Comment 1**
> The organization of the paper could be further improved. For example, the definition of 6D rotations is only explained in Sec 1.2. It would be better to mention it the first time this term appeared, otherwise, it is slightly confusing. Besides, it is suggested to merge the five contributions into three or four. e.g. merge the fourth and fifth contributions.
>
> **Response 1**
> As suggested by the reviewer, we now define a 6D rotation representation that we employ the first time it is mentioned and we merge the fourth and fifth contributions.
>
> **Comment 2**
> Some important details are missing. a). How are the metrics L^{det}{gpd}, L^{det}{ikd}, L^{det}_{loc} defined and what are the units of them? b). How are the two datasets split into train and evaluation?
>
> **Response 2**
>
> a) Indeed, we have moved the definitions of L^{det}{gpd-L2}, L^{det}{ikd-L2} and L^{det}{loc-geo} to appendices, together with Algorithm 1, without noticing that this caused some loss of clarity in the main manuscript. We thank the Reviewer for pointing this out. In the revised manuscript, we have added explicit definitions of the metrics to the Appendix E and referenced it in Section 3.2 while defining the training and evaluation setup. L^{det}{gpd-L2}, L^{det}{ikd-L2} are mean squared errors, therefore their units are in meters squared and L^{det}_{loc-geo} is the geodesic error, measured in radians.
>
> b) The dataset splitting procedure is described in Section 3.1: Datasets. In particular, for miniMixamo it is defined as “The resulting dataset is partitioned at the clip level into train/validation/test splits (with proportion 0.8/0.1/0.1,respectively) by sampling clip IDs uniformly at random. Splitting by clip makes the evaluation framework more realistic and less prone to overfitting: frames belonging to the same clip are often similar. At last, the final splits retain only 10% of randomly sampled frames (miniMixamo has 33,676 frames total after subsampling) and all the clip identification information (clip ID, meta-data/description, character information, etc.) is discarded.” For miniAnonymous the splitting procedure is the same as per “Then we create a dataset of a total of 96,666 subsampled frames following exactly the same methodology that was employed for miniMixamo.”

---

### Official Review · Reviewer_J8Sg · 2021-11-02

**Correctness:** 3
**Technical Novelty And Significance:** 3
**Empirical Novelty And Significance:** 3
**Recommendation:** 8
**Confidence:** 3

**Details Of Ethics Concerns:**

I have reviewed this paper and I think there are no ethical concerns.

**Main Review:**

Pros:

(1) This paper focuses on an interesting and important task in human animation. The proposed method and animation tool may alleviate the workload of animation by the artists.

(2) Several modifications about the model architectures are clearly described, and most of them make sense to me.

(3) This paper provides comprehensive materials, including the experiments, appendix, and demo videos. The effectiveness of most of the proposed modules is proved by the experiments.

Cons:

(1) The proposed method is a new way for human animation, so the comparison with FinalIK should be more detailed.
The speed of both methods should be clearly shown in Table 1.

As mentioned in the paper,  the proposed method satisfies the constraints approximately (P.9 The limitation of the current work). The construction results for both methods with full constraints may help to clarify the limitation.

The generalization ability of both methods should also be investigated. How about performing both methods on non-human skeletons with similar structures.

(2) Although this paper has provided very comprehensive results, some comparisons and analyses should be further shown in the experiment part.

In Table 1, the inference speed & parameter numbers should be provided for machine learning-based methods.

Quantitative evaluation for the look-at-loss should be presented to prove the effectiveness.

Table 2 shows the ablation study for GPD, but how about only using GPD (without IKD) for the final result.

(3) The dataset is one of the most important contributions in this paper. It is better to show more details about the dataset, especially the 'miniAnonymous' dataset.

How many instances are included in the dataset?

What kinds of actions for each instance are captured?

Does the motion style have a big difference with the 'miniMixamo' dataset?

(4) Some descriptions of the proposed unity tool should be presented. Does it just show the results of the neural network? Or some animation functions in the unity is also used as post-processings?

Detailed questions:

(1) What is the minimal number of the input constraints?

(2) What kind of results will be output if a sequence of the ambiguous input constraints is given?

(3) There are many interesting comparisons in the demo videos, so it is better to show some of them in the main paper.

(4) In Table 5 and Table 6, there are some distinct results, i.e. 1.27 e-6. Are those typoes?

**Summary Of The Paper:**

This paper focuses on the learnable neural representation of the 3D human pose. A proto-residual network is proposed to construct a full human pose from sparse body part constraints. Several modifications are performed to improve the accuracy of the proposed model, such as look-at loss, two-stage decoder, residual scheme, etc. In the experiment, two new datasets for the static human pose modeling task are conducted and will be released publicly. The experimental results prove the effectiveness of the proposed methods.

**Summary Of The Review:**

This paper focuses on a very interesting topic and proposes two datasets that may facilitate the research area. The proposed methods are reasonable, and most of them are well evaluated.  Although some experiments should be added and some details are missing now,  I think it is still a good submission and ok to be accepted.

---

> ### Author Response · Authors · 2021-11-18
> **Author response to Reviewer J8Sg 4/4**
>
> **Comment 11**
> Does the motion style have a big difference with the 'miniMixamo' dataset?
>
> **Response 11**
> In Appendix D, we have included the histograms summarizing top 30 tags appearing in the descriptions of the individual frames that comprise both datasets. The analysis of the histograms shows that there is significant difference in the styles of motion captured in the datasets. The following text has been added to Appendix D. “The distributions of 30 most popular tags across the frames of the original dataset used to build miniMixamo and miniAnonymous are shown in Figure 5. It is clear that the two datasets cover some common activities such as walking and idling, for example. Additionally, there are numerous categories the two datasets emphasize separately. For example, miniMixamo focuses a lot on fighting and handling guns, whereas miniAnonymous has more neutral energy activities, provides extensive labeling of male/female poses as well as covers scenarios of handling various objects and food.” Tables 6, 7 provide additional information about motion types in both datasets. Particularly, Table 6 describing our dataset provides very fine grained specification of motion types based on our data collection protocol design.
>
> **Comment 12**
> (4) Some descriptions of the proposed unity tool should be presented. Does it just show the results of the neural network? Or some animation functions in the unity is also used as post-processings?
>
> **Response 12**
> We thank the reviewer for the insightful comment. The Unity tool that is used to produce demo videos and qualitative pictures show the **raw neural network outputs without applying any post-processing**. This is done to ensure fair comparison of different methods and avoid any misleading results due to post-processing. We have clarified this in the manuscript when discussing our demos.
>
> **Comment 13**
> (1) What is the minimal number of the input constraints?
>
> **Response 13**
> The minimal number of constraints is one.
>
> **Comment 14**
> (2) What kind of results will be output if a sequence of the ambiguous input constraints is given?
>
> **Response 14**
> In a sense, all our results are based on ambiguous constraints, because we test the system based on sparse effector inputs, which provide very little information about the final pose. Our results indicate that the learned IK model is able to successfully reconstruct plausible poses using minimal inputs. For example, Figure 3 demonstrates how pose can be designed adding progressively more effector inputs. The pose guided with a small number of effectors (e.g. two effectors in the case of Figure 3, left) tends to be very generic, picking up a prominent pattern from the underlying dataset. Adding effectors (moving left to right in Figure 3) helps make the pose more and more specific. In general, our demos show that the learned pose manifold is reasonably smooth and stable: small perturbations in effectors result in small perturbations of the resulting pose.
>
> **Comment 15**
> (3) There are many interesting comparisons in the demo videos, so it is better to show some of them in the main paper.
>
> **Response 15**
> We would like to thank the Reviewer for their positive assessment of our demo videos. Unfortunately, this year there is no extra page provided for addressing the comments. The current manuscript has already undergone a few compacting iterations in the attempt to strike the balance between being well-motivated, self-contained from the technical perspective as well as providing some intuitive visuals and supporting claims with strong quantitative evidence. Unfortunately, we have not been able to identify the pieces in the current manuscript that could be removed to provide space for additional qualitative results. We will be more than happy to look at this issue again if Reviewer J8Sg provides additional opinion on which parts of the main body could be removed to accommodate the new material, without compromising the clarity of the manuscript.
>
> **Comment 16**
> (4) In Table 5 and Table 6, there are some distinct results, i.e. 1.27 e-6. Are those typoes?
>
> **Response 16**
> We sincerely thank the Reviewer for pointing out the typos. We have reviewed the tables and fixed the typos in the current revision of the paper.

---

> ### Author Response · Authors · 2021-11-18
> **Author response to Reviewer J8Sg 3/4**
>
> **Comment 4**
> (2) Although this paper has provided very comprehensive results, some comparisons and analyses should be further shown in the experiment part.
>
> **Response 4**
> We thank Reviewer J8Sg for their positive assessment of our work and provide detailed responses below.
>
> **Comment 5**
> In Table 1, the inference speed & parameter numbers should be provided for machine learning-based methods.
>
> **Response 5**
> In the revised manuscript we have provided the inference speed and and parameter counts in Table 1, as reviewer suggested. In particular, the inference speed of FinalIK on CPU is 0.3ms and the inference speed of ProtoRes on GPU is around 5.5 ms and on CPU 13.5 ms. Our setup for the pose authoring inference scenario is NVIDIA Geforce RTX 2080 Super, Intel Core i7 2.30 Ghz.
>
> **Comment 6**
> Quantitative evaluation for the look-at-loss should be presented to prove the effectiveness.
>
> **Response 6**
> In our training and evaluation pipeline, we currently log only the losses currently presented in the manuscript. Therefore, collecting the metrics requested by Reviewer J8Sg will require additional code modifications and rerunning the training sessions. We will include the look-at-loss results in the final version of the revised manuscript. In the meanwhile, our qualitative results provide evidence to show the effectiveness of the proposed look-at loss.
>
> **Comment 7**
> Table 2 shows the ablation study for GPD, but how about only using GPD (without IKD) for the final result.
>
> **Response 7**
> Using GPD without IKD is not feasible, because GPD predicts joint positions without respecting skeleton constraints, such as bone lengths, therefore its output cannot be used for pose design directly. Additionally, positions alone are not sufficient to model bone twisting around its axis, which is desirable in the upstream applications. Finally, the task of predicting only joint positions is significantly easier than the task of predicting full pose satisfying skeleton constraints, so results would not even be directly comparable and rotation output would be missing for rotation metrics computation.
>
> **Comment 8**
> (3) The dataset is one of the most important contributions in this paper. It is better to show more details about the dataset, especially the 'miniAnonymous' dataset.
>
> **Response 8**
> We thank the reviewer for pointing this out, we have addressed the concern by extending Appendix D with additional information about datasets, including some descriptive statistics and more detailed description of the contents. Detailed responses to related questions follow.
>
> **Comment 9**
> How many instances are included in the dataset?
>
> **Response 9**
> We are uncertain about what is meant by ‘instances’. We list here some potentially relevant information, but we are happy to expand on other aspects given clarifications on the term ‘instance’.
>
> The miniMixamo dataset contains 33 676 poses, and the miniAnonymous dataset contains 96 666 poses. The miniMixamo dataset contains poses from 1598 clips, while the miniAnonymous dataset contains poses from 1776 separate clips. Note that these details are available in Section 3.1 of the manuscript.
>
> **Comment 10**
> What kinds of actions for each instance are captured?
>
> **Response 10**
> We have added the hierarchy of action styles and categories captured in the miniAnonymous and miniMIxamo datasets in Appendix D, especially Tables 6, 7. The following text has been added to address the concern: “The following action scenarios were used to collect MOCAP sequences in miniAnonymous. First, the following locomotion types: compass crouch, compass jogs, compass runs, compass walks were collected for female and male subjects under high energy, low energy and injured scenarios. The same locomotion types were collected under neutral energy feminine and neutral energy macho scenarios. Furthermore, under neutral energy generic scenario, for both female and male subjects, we collected following action, object and environment interaction types: archery, bokken fighting, calisthetics, door interactions, fist fighting, food, handgun, hands, knife fighting, locomotion, longsword fighting, phone, place, railing interactions, rifle, seated interactions, shotgun, standings, sword, wall. Among the latter categories, locomotion and handgun had following more detailed subdivisions. Locomotion: compass crawls, compass crouch, compass jogs, compass rifle aim walks, compass rifle crawl, compass rifle crouch, compass rifle jogs, compass rifle runs, compass rifle walks, compass runs, compass walks, rifle idles, walk carry heavy backpack, walk carry heavy sack, walk carry ladder, walk dragging heavy object. Handgun: downwards, level, upwards, verticals.”

---

> > ### Comment · Reviewer_J8Sg · 2021-11-22
> > **Clarification about comment 9**
> >
> > Thanks for the detailed feedback. The miniAnonymous dataset is captured by a MOCAP studio, so I'd like to ask how many persons are captured for the dataset?

---

> > > ### Author Response · Authors · 2021-11-23
> > > **Re: Clarification about comment 9**
> > >
> > > We thank Reviewer J8Sg for additional question. The dataset captures two people, one male and one female, both are professional performers, capable of delivering high dynamic range movements in a variety of scenarios, whose precise specifications were provided in the data collection protocol (see Table 6 in Appendix D). The MOCAP data from both subjects were retargeted to a standardized skeleton, which is a standard practice for MOCAP data collection. Our dataset achieves the following objectives that we set forth for our data collection exercise:
> > > - Collect data that have balanced coverage of male and female poses
> > > - Cover actions in a wide dynamic range of energy (low, neutral, high, injured)
> > > - Cover diverse variety of activities
> > >
> > > We will clarify it in the revised version of the manuscript accordingly.

---

> > > > ### Comment · Reviewer_J8Sg · 2021-12-01
> > > > **Re:Re: Clarification about comment 9**
> > > >
> > > > Thanks for the feedback. It is clear to me now.

---

> ### Author Response · Authors · 2021-11-18
> **Author response to Reviewer J8Sg 2/4**
>
> **Comment 2**
> As mentioned in the paper, the proposed method satisfies the constraints approximately (P.9 The limitation of the current work). The construction results for both methods with full constraints may help to clarify the limitation.
>
> **Response 2**
> We believe that our demo video 4_Final_IK_comparison.mp4 already reveals the limitation qualitatively. To better focus the reader’s attention on the qualitative results in this demo, while discussing the limitation in detail, we have updated the text discussing the limitation in Appendix L, Limitations as follows.
>
> “Constraints are not satisfied exactly, as opposed to the conventional systems. The limitation can be observed in the demo video 4_Final_IK_comparison.mp4, which compares ProtoRes and FinalIK side-by-side. For example, at seconds 18-21 we can see the chest position is not satisfied exactly by ProtoRes and the generated left foot position traverses the vicinity of the constraint as the user changes the position of the left foot. Another example like this can be seen at seconds 35-41, in which the left foot effector is not satisfied exactly by ProtoRes. It is interesting that for the rather peculiar configuration of effectors, FinalIK produces outputs that tend to severely twist the joints, perhaps beyond the capabilities of an average human. At the same time, ProtoRes tends to trade the precision of following the individual effectors in isolation with the plausibility of the overall pose. This is the price to pay for the ability of the model to inject the data-driven inductive bias that can be used to reconstruct pose from very sparse inputs. This could be mitigated using a conventional solver on top of the trained model. In this case, the model would produce a globally plausible pose, whereas the solver could only do the final pass to strictly satisfy certain constraints. Also, to provide additional flexibility in solving some of the constraints more strictly than the others, our model provides an effector tolerance mechanism that can help the user trade off the strictness of satisfying certain effectors vs. some others.”
>
> **Comment 3**
> The generalization ability of both methods should also be investigated. How about performing both methods on non-human skeletons with similar structures.
>
> **Response 3**
> Unfortunately, the detailed investigation of the generalization ability of FinalIK is outside of the scope of our current paper as it is not our contribution. Generally speaking, FinalIK allows manually defining custom effectors and custom IK chains to accommodate for new skeletons via a tedious hand crafted process. On the other hand, the generalization ability of our core contribution ProtoRes has been investigated in the current work. We provide the demo video 7_Quadruped.mp4 and associated Appendix K.7 describing it, in which we demonstrate the results of learning a neural representation of the dog skeleton using ProtoRes. Note that we did not tune any hyperparameters or change our model training code beyond the dataset loader part to accomodate for the new data source. The demo shows successful learning results with the quadruped dataset supporting the same flexibility/realism and showcasing similar features as our human model. In our view, this provides evidence to support the promising generalization potential of our model beyond human skeletons.

---

> > ### Comment · Reviewer_J8Sg · 2021-11-22
> > **About the comparison of the proposed method and FinalIK**
> >
> > Thanks for the author's feedback. It is better to show the quantitative results of both methods with full constraints for clarification.

---

> > > ### Author Response · Authors · 2021-11-23
> > > **FinalIK does not support full constraints**
> > >
> > > We thank Reviewer J8Sg for additional clarifications. We would like to ask the reviewer to provide extra feedback based on the additional pieces of information we include below.
> > >
> > > - First, we are not completely clear about the exact definition of the “full constraints” scenario. If this is the scenario in which the model is given positions and rotations of all joints, a non-learnable inverse kinematics solution capable of doing IK on all joints will produce the output with zero error, **provided that the pose is feasible** (i.e. bone lengths are respected). Essentially, the IK system in this case would deliver an identity transform, subject to arithmetic round-off errors. This implies that if such a system could be available, measuring its performance would not even be necessary. It would only be of interest to measure the accuracy of ProtoRes operating on a fully specified pose input.
> > > - Second, the “full constraints” experiment comparing FinalIK against ProtoRes is not feasible and we cannot possibly implement it, because by design FinalIK does not support the mode of operation with full constraints. This is a limitation of the system outlined in its documentation: http://www.root-motion.com/finalikdox/html/page8.html. We present a few excerpts from the documentation supporting the statement for the convenience of the reader. “FullBodyBipedIK does not have effectors for the fingers and toes… Using 10 4-segment constrained CCD or FABRIK chains to position the fingers however is probably something you don't want to waste your precious milliseconds on.”; “FullBodyBipedIK does not have elbow/knee effectors.”; “FullBodyBipedIK does not rotate the shoulder bone when the character is pulled by the hand.” Additionally, the actual set of potential effectors provided by the package in the Unity environment is even more limited. Realistically, the configuration available with FinalIK package provides us with at most 13 effectors total, including positional: body trunk (chest point), 2 hands, 2 feet, 2 shoulders, 2 thighs and rotational: 2 hands, 2 feet.
> > > - Third, we would like to point out that the aforementioned limitations of FinalIK stem from the typical use cases (i.e. resolution of sparse inputs) in which IK systems tend to be used. The underlying intent behind IK systems is to reduce the number of user inputs required to produce a believable pose. Therefore, the operation with "full constraints" seems to represent a use-case neither for FinalIK nor for ProtoRes.

---

> > > > ### Comment · Reviewer_J8Sg · 2021-12-01
> > > > **Feedback to the full constraints issue**
> > > >
> > > > 1. Full constraints.
> > > > Yes, it is the scenario where the positions and rotations of all joints are given.
> > > >
> > > > 2. The results for full constraints.
> > > > I agree that the full constraints case is not a typical case for IK, and in that case, the final IK may generate a very accurate result. But I still think the quantitative result of the proposed protoRes method with full constraints is meaningful. The result may help the readers to learn more about the limitations of the proposed method.

---

> > > > > ### Author Response · Authors · 2021-12-01
> > > > > **Re: Feedback to the full constraints issue**
> > > > >
> > > > > We would like to thank Reviewer J8Sg for the additional feedback! To address the point raised by the reviewer, we will additionally report the quantitative result for ProtoRes in the full constraints scenario (the positions and rotations of all joints are given) in the supplementary of the revised manuscript.

---

> ### Author Response · Authors · 2021-11-18
> **Author response to Reviewer J8Sg 1/4**
>
> We would like to sincerely thank **Reviewer J8Sg** for careful reading of the manuscript and a positive and encouraging review of our work. We address the points raised by the Reviewer in detail below. Please do not hesitate to post additional comments and questions, we will be more than happy to address them.
>
> **Comment 1**
> (1) The proposed method is a new way for human animation, so the comparison with FinalIK should be more detailed. The speed of both methods should be clearly shown in Table 1.
>
> **Response 1**
> To address the comment, we have added the quantitative inference speed comparison for the two methods to Table 1, as suggested by the reviewer. Our setup for the pose authoring inference scenario is NVIDIA Geforce RTX 2080 Super, Intel Core i7 2.30 Ghz. ProtRes inference time per pose solve, 5 positional effectors + 1 head lookat scenario, CPU (proprietary CPU based accelerator engine, multi-threaded with Burst compilation): 13.5 ms; GPU (OnnxRuntime): 5.5 ms. FinalIK inference time per pose solve, 5 positional effectors for full-body constraint + 1 head for look-at constraint scenario, CPU (single-threaded): 0.3 ms. Additionally, based on our practical use of the integrated system, we observe that ProtoRes runs at least at 100 fps speed, making its use in the context of the pose editing tool smooth and seamless: the user of the system does not experience any delays that could be caused by neural network computations. This is also obvious from the demo video 4_Final_IK_comparison.mp4, which was shot in real time, and compares ProtoRes and FinalIK side-by-side. In this video, we do not observe any delays on the ProtoRes side when running both methods at the same time in the same setup in real time. Our conclusion is that for the pose authoring applications ProtoRes offers a computationally viable solution that can be run seamlessly on both CPU and GPU.

---

### Official Review · Reviewer_cvbx · 2021-11-02

**Correctness:** 3
**Technical Novelty And Significance:** 3
**Empirical Novelty And Significance:** 2
**Recommendation:** 8
**Confidence:** 4

**Main Review:**

**Strength**

1. **Impressive demo and usable system:** this work focuses on a practical and important problem: inverse kinematics based on a sparse set of user-specified constraints, and builds a usable system in Unity that shows impressive results in generating realistic and natural-looking human poses based on fast-changing inputs.  The proposed system looks complete and deployable for graphics and animation use-cases.
2. **Proposed network:** the use of prototypical networks in the task of human pose generation or pose infilling, to the best of my knowledge, is novel, though it would be beneficial if the authors can provide more in-depth motivation and in-depth analysis of why such network design is more suitable in the proposed task. What inductive bias is prototypical networks leverage in this specific task?
3. **Comprehensive ablation:** the ablation studies provided in the paper and supplement substantiate the benefits of the proposed learning procedure and loss weighting schemes.

**Weakness**

1. **Missing simple baselines:** based on the problem setup, one simple baseline is to use nearest neighbors to retrieve the closest pose from the dataset that satisfies the given input. Since MoCap data is continuous, it is highly possible that some realistic pose already exists in a large enough human motion database (such as AMASS [3], around 3 million frames). If no exact match exists, another naive approach could be finding close samples and interpolating between them either through simple linear interpolation or learned embeddings such as VPoser [4].
2. **Missing motivation and analysis for the proposed "ProtoRes" architecture**: as mentioned in the strength section, it is interesting the prototypical networks are used for this task. However, it is not well-motivated and not fully studied why the proposed network architecture is chosen for this task. Given the recent progress in Transformers, it is possible that the Transformer requires more data (which a dataset like AMASS can provide) but can learn a better embedding space for the pose synthesis/infilling task.
3. **Missing generative aspect:** for such an under-constrained problem, it is natural to assume that generative models such as normalizing flow [5] or VAE [6] are a more suitable model for generating diverse pose that conforms to the same user input.

**Minor Issues**

1. **Multi-task learning:** I am not sure if multi-task learning is the right term for describing the loss terms: the look-at, rotation, position loss for joints are largely the same objective expressed in different modalities. Since all loss is calculated in a root-relative coordinate system, the rotation term should contain all information available to reconstruct 3D positions through forwarding kinematics. Using both rotation and positions more likely served as a more explicit representation of the objective. This objective has also been utilized in some recent works in pose estimation [1][2].
2. **Variable input length:** while variable input length is mentioned as a feature of the proposed system, it is not demonstrated via demo or experiments. Since there is only a fixed number of joints, I presume the system can use a fixed number of inputs (all the joints) and use a mask to indicate whether there is an input for that specific joint?


==================================================


**After author response**

After reading the author's response and other reviews, most of my concerns are resolved. Thus, I would like to raise my score to 8 for this work for its novel use of prototypical networks in human pose modeling and the practical use case of the results.


[1] Li, Jiefeng, Chao Xu, Zhicun Chen, Siyuan Bian, Lixin Yang and Cewu Lu. “HybrIK: A Hybrid Analytical-Neural Inverse Kinematics Solution for 3D Human Pose and Shape Estimation.” CVPR 2021

[2] Rempe, Davis, Tolga Birdal, Aaron Hertzmann, Jimei Yang, Srinath Sridhar and Leonidas J. Guibas. “HuMoR: 3D Human Motion Model for Robust Pose Estimation.” ICCV 2021

[3] Mahmood, Naureen, Nima Ghorbani, Nikolaus F. Troje, Gerard Pons-Moll and Michael J. Black. “AMASS: Archive of Motion Capture As Surface Shapes.” *CVPR 2019*

[4] Pavlakos, Georgios, Vasileios Choutas, Nima Ghorbani, Timo Bolkart, Ahmed A. A. Osman,

Dimitrios Tzionas and Michael J. Black. “Expressive Body Capture: 3D Hands, Face, and Body From a Single Image.” CVPR 2019

[5] Henter, Gustav Eje, Simon Alexanderson and Jonas Beskow. “MoGlow: Probabilistic and controllable motion synthesis using normalising flows.” *ACM TOG, 2020*

[6] Yuan, Ye and Kris Kitani. “Diverse Trajectory Forecasting with Determinantal Point Processes.” ICLR 2019

**Summary Of The Paper:**

This paper develops a system to generate a complete and realistic human pose based on a sparse set of inputs such as look-at direction and end-effector positions. It proposes a prototypical residual network dubbed " ProtoRes" to handle variable input length based on the user-specified parameters. A look-at-loss, randomized weighting scheme, and two datasets based on MoCap are proposed to learn the designed system, and a usable system for authoring realistic human poses is built. Experiments show that the network design and loss outperform baselines such as Transformer and existing Inverse Kinematics methods. The graphical demo built inside Unity also showcases the utility of the learned pose networks.

**Summary Of The Review:**

Overall, the proposed system is a practical and usable system for pose authoring based on sparse input. The demo is impressive and seems a great addition to inverse kinematic systems. On the other hand, the technical novelty seems lacking and some important baselines are missing. In all, I think this work is borderline and would like to hear the author's response before making the final decision.

---

> ### Author Response · Authors · 2021-11-11
> **Author response to Reviewer cvbx 5/5**
>
> **Comment 5**
> Multi-task learning: I am not sure if multi-task learning is the right term for describing the loss terms: the look-at, rotation, position loss for joints are largely the same objective expressed in different modalities. Since all loss is calculated in a root-relative coordinate system, the rotation term should contain all information available to reconstruct 3D positions through forwarding kinematics. Using both rotation and positions more likely served as a more explicit representation of the objective. This objective has also been utilized in some recent works in pose estimation [1][2].
>
> **Response 5**
> We would like to thank the reviewer for the insightful comment. The definition of multi-task learning is generally pretty broad - this is a multi-faceted domain. Based on Caruana 1997, the common features reminiscent of multi-task learning are as follows: shared input, shared representation, concurrent learning on multiple tasks, each task corresponds to a different neural network output and a different objective, concurrent learning on multiple tasks improves generalization by providing a stronger inductive bias. “Inductive bias is anything that causes an inductive learner to prefer some hypotheses over other hypotheses.” (Caruana 1997). We believe all these features are present in our setup. Our network has the same input, it has a shared representation, it has several outputs: local rotations and positions after an FK pass are good examples, the network is trained concurrently on several objectives, including local rotation loss and the global position loss, among others; we show that training on the global position and local rotation losses concurrently, improves generalization on both of them. The latter fact, in our view, confirms that the position- and rotation-related internal network information flows are genuinely different. It appears that solving positions and rotations requires the network to exercise slightly different inductive biases that, when combined together, produce a better overall result.  We feel that the Reviewer’s argument “Using both rotation and positions more likely served as a more explicit representation of the objective.'' does not actually contradict the multi-task scenario definition from Caruana 1997. We feel this is very much aligned with the notion of inductive bias, i.e. the ability of the learner to prefer some hypotheses over others, in turn making the overall objective easier to reach, or in other words making it more explicit. Indeed, we show both quantitatively and qualitatively that concurrent training on rotation and position losses improves generalization on both of them (cf lines 1, 6, 7 in Table 2). Note that since we are talking about generalization performance on the test set, it is clear that the network accumulates generalizable insights (inductive biases) helping it better solve unseen sparsely defined poses. Our qualitative example video 3_Loss_Ablation.mp4 demonstrates that this is not a coincidence. When rotation loss is removed, the network is able to solve the joint positions well, however rotations are totally off, making the overall pose implausible. With the full loss term including rotations, the network learns the inductive bias of choosing poses that respect both positions and rotations, making the overall pose qualitatively more appealing, which also shows quantitatively in improved position and rotation metric scores (cf lines 1 and 6 in Table 2). Finally, although [1,2] use a variety of loss terms, including a combination of position and rotation losses, they do not show that using a combination of rotation and position terms is critical for recovering a plausible full pose from a sparsely defined one. Our contribution in this domain is therefore two-fold: (i) we present qualitative and quantitative evidence implying that using both rotation and position is beneficial for recovering plausible poses from sparse variable user inputs and (ii) we introduce the randomized loss weighting scheme, which appears to be necessary to achieve synergistic effects while concurrently training on position and rotation losses, as follows from our Table 2.
>
> (Caruana 1997) Caruana, R. Multitask Learning. Machine Learning 28, 41–75 (1997)

---

> ### Author Response · Authors · 2021-11-11
> **Author response to Reviewer cvbx 4/5**
>
> **Comment 4**
> Missing generative aspect: for such an under-constrained problem, it is natural to assume that generative models such as normalizing flow [5] or VAE [6] are a more suitable model for generating diverse pose that conforms to the same user input.
>
> **Response 4**
> We thank the reviewer for pointing this out. Indeed, generative/distributional modeling sounds like a very interesting venue for future research. A few detailed points follow.
>
> - We would like to point out that architecturally, our approach can be modified straightforwardly into a generative model by predicting $\mu$ and $\sigma$ from the pose embedding, incorporating the KL divergence term and decoding pose from the associated latent variable, for example. However, there are a couple of problems that need a significant amount of work to make the generative approach viable and useful in our application scenario.
> - First and foremost, we need to develop effective user interface tools which would allow the user to explore the random outputs that the generative model might be able to produce without overwhelming the user with noise. We need to understand how to best present the noisy results to the user (density maps, a few potential poses, one pose per click etc.) and how to enable the user to navigate/select generated pose samples. This is an exciting and non-trivial problem in itself. Unfortunately, until it is solved, the practical use of generative models for our specific problem is very limited. The focus of our current work is to show that the problem of producing realistic human poses using a small set of heterogeneous variable user inputs has a computationally efficient, elegant and practical solution via learned inverse kinematics and neural modeling approaches. We believe that our results validate this and open up opportunities for future research, including generative modeling of partially defined human poses.
> - Second, the performance of generative models would need to be benchmarked to assess the quality of distributional modeling/predictions and provide an objective benchmark to support future model development in this area. Many existing works focus on demonstrating the prediction accuracy of generative models or evaluating the likelihood values of their predictions. We believe that more work needs to be done in this direction to make the assessment of generative models more thorough and objective. For example, high accuracy can be achieved by approaches that do not rely on generative modeling. Likelihood value assessment is not ideal either: the model that only generates samples with high likelihood (and scores well) basically is bound to produce only the most common values, which may not be attractive from the user perspective. The model that produces too many samples in the tails of distribution might not be good either, because it will be hard to steer by the user and will produce too many samples that do not fit the user's guidance signals. More research needs to be done in this direction to identify suitable ways of quantifying the diversity of samples, the adequacy of the overall distribution fit as well as the usefulness of the randomly generated samples for the user of the system. We believe this is a very exciting area for future research.

---

> ### Author Response · Authors · 2021-11-11
> **Author response to Reviewer cvbx 3/5**
>
> **Comment 3**
> Missing motivation and analysis for the proposed "ProtoRes" architecture: as mentioned in the strength section, it is interesting the prototypical networks are used for this task. However, it is not well-motivated and not fully studied why the proposed network architecture is chosen for this task. Given the recent progress in Transformers, it is possible that the Transformer requires more data (which a dataset like AMASS can provide) but can learn a better embedding space for the pose synthesis/infilling task.
>
> **Response 3**
> The initial motivation for using the prototypical network for this task was based on the fact the prototypical networks had been shown to be effective at encoding sparse semantic information. Since in our task we have only sparse pieces of information, it felt logical to pursue this powerful and simple approach. We have outlined this in the introduction. Furthermore, Section 2 further details the motivation in the discussion following equations (2)-(5). First, the prototypical approach is used to represent the pose defined by sparse effectors, which is a use case resonating well with a few-shot learning scenario and implementing the inductive bias that when a pose is specified by a few effectors, the mean of their representations is the most concise representation of the pose. We also show that extending such an approach using residual stacking methodology results in significant accuracy improvements in our problem. Second, in each block the representation of the entire pose is subtracted from the representations of individual effectors, implementing the inductive bias that the information in individual effectors is only valuable when it is different from what is already stored in the embedding of the entire pose. Finally, pose representation is accumulated across residual blocks (prototype across block representations) effectively implementing skip connections from very early layers. This is a computationally effective way of implementing distant skip connections. Empirically, we show that our approach is superior with respect to Transformer, both computationally and in terms of accuracy on two different datasets. In our view this is a significant contribution given our goal of democratizing AI assisted artistry and making associated tools accessible to everyone, whereas the Transformer based approach might be computationally prohibitive (10 times training cost compared to ProtoRes in our setup), especially with larger datasets. Moreover, we ran extensive ablation experiments studying different aspects of the proposed architecture, including, among other things, the experiments showing the advantages of the proposed  Prototype-Subtract-Accumulate residual architecture as well as the computationally efficient encoder/decoder design.

---

> ### Author Response · Authors · 2021-11-11
> **Author response to Reviewer cvbx 2/5**
>
> **Comment 2**
> Missing simple baselines: based on the problem setup, one simple baseline is to use nearest neighbors to retrieve the closest pose from the dataset that satisfies the given input. Since MoCap data is continuous, it is highly possible that some realistic pose already exists in a large enough human motion database (such as AMASS [3], around 3 million frames). If no exact match exists, another naive approach could be finding close samples and interpolating between them either through simple linear interpolation or learned embeddings such as VPoser [4].
>
> **Response 2**
> We thank the reviewer for the insightful comment and provide a detailed response below.
>
> - We would like to note that in our work we are looking for strong generalization, therefore the dataset is split into train/validation/test by sampling sequence ids first. Thus frames from the same sequence can only be part of one of the splits. Therefore, the generalization ability of the nearest neighbor approach will not be as high as one might expect if we could rely on the continuous nature of MoCap data and adjacent frames occuring in train and test splits.
> - The nearest neighbor approach is popular in motion completion and interpolation. In this case the notion of “exact match” is straightforward because (1) inputs are always the same, and (2) the frame of reference is always defined in the same way (e.g. root transform of the current frame). In our case, we solve a more challenging problem, in which any arbitrary combination of joints can be used as inputs. This makes the frame of reference ambiguous (e.g. if we don't use the root effector's full transform) and changing for every effector combination, making the “exact match” very hard to define for all possible effector combinations.
> - Moreover, the efficient nearest neighbor inference algorithm based on kd-tree would require a construction of the tree for each viable combination of input joints, implying combinatorial complexity. Similarly, VPoser [4] only models a fixed size input using fully connected MLP (please refer to Sections 3.1, 3.3 and Supplementary Section 8 in [4]). As such, a latent space learned by VPoser is not applicable for solving our problem. If we chose to use it naively and learn a separate model for every possible combination of inputs, we will face the same problem as the nearest neighbor, i.e. we will have to fit a combinatorial number of VPoser models for all possible input joint combinations.
> - We additionally handle the complexity of solving the problem with heterogeneous inputs: position, rotation, look-at, in arbitrary combinations and numbers. If we try to implement this using the nearest neighbor approach, we will have to face a few more problems. First, we will have to assign weights to position, rotation and look-at distance metrics and run grid search to figure out the best configuration of weights for each distance term. To achieve this, we will have to run a computationally involved grid search procedure to identify the optimal weight for each distance term. Second, since the distance term for this problem will be a custom composite function, we will have to work on a custom implementation of the kd-tree (see e.g. https://stackoverflow.com/questions/48042408/kd-tree-with-custom-distance-metric). Third, even if we code it successfully and resolve the inference speed problems, we will still have to face the combinatorial expansion of model space, which will be even bigger when we expand the space of possibilities with effector type in addition to the arbitrary joints.
>
> To sum up, we argue that the baselines proposed by Reviewer cvbx are neither simple nor feasible in order to solve the full extent of our problem. VPoser [4] is not applicable to solve our problem, because it can only work with the fixed inputs covering the entire set of joints of the body and it can only handle positional inputs. We believe that the baselines we present in the paper correspond well to the complexity of the task we are solving and adequately represent the existing technology that can be straightforwardly applied to solve it.

---

> > ### Comment · Reviewer_cvbx · 2021-11-21
> > **Clarification about VPoser**
> >
> > I thank the authors for the detailed response. Some clarification about VPoser: I was suggesting using it as a way to interpolate between close samples based on nearest neighbors, not as the tool for finding samples on its own. I think the point about the variable number of inputs and reference frame difficulty is valid.

---

> ### Author Response · Authors · 2021-11-11
> **Author response to Reviewer cvbx 1/5**
>
> We would like to sincerely thank **Reviewer cvbx** for careful reading of the manuscript and a constructive and encouraging review of our work. We address the points raised by the Reviewer in detail below. Please do not hesitate to post additional comments and questions, we will be more than happy to address them.
>
> **Comment 1**
> Variable input length: while variable input length is mentioned as a feature of the proposed system, it is not demonstrated via demo or experiments. Since there is only a fixed number of joints, I presume the system can use a fixed number of inputs (all the joints) and use a mask to indicate whether there is an input for that specific joint?
>
> **Response 1**
> We would like to point out that the variable input length functionality is demonstrated via both demos and experiments.
>
> - Our architecture natively supports variable inputs of various types
> - Figure 3 provides qualitative demonstration of the poses that can be achieved using 2,4,5,7,4,7 effectors (left to right) of different types (position, look-at, rotation)
> - Demo video 1_ProtoRes_Demo.mp4 demonstrates that the effectors can be added and removed freely by the user, showing posing results with variable number and types of input effectors. To clarify, our ProtoRes model does not rely on masking of inactive effectors to process inputs, but performs parallel processing of active effectors, and combines them into a fixed-size pose embedding through the Prototype-Subtract-Accumulate process.
> - Our empirical results are based on the random evaluation framework, testing variable number of input effectors and variable effector types (see Section 3.2 and Appendix E). “The evaluation framework tests model performance on a pre-generated set of seven files containing 6,7...12 effectors respectively. Metrics are averaged over all files, assessing the overall quality of pose reconstruction in a scenario with sparse variable inputs.”
>
> Additionally, we have used the approach in which the missing outputs are masked with learnable placeholders as a baseline. This is exactly what the Maksed-FCR baseline does (see Section 3.3, first paragraph) “The first baseline, Masked-FCR, is a brute-force unstructured baseline that uses a very wide J · 3 · 7 input layer (J joints, 3 effector types, 6D effector data and 1D tolerance) handling all effector permutations. Missing effectors are masked with 3 · J learnable 7D placeholders. MaskedFCR has 3 encoder and 6 decoder blocks to match ProtoRes.” Our results in Table 1 clearly indicate that this approach is inferior to ProtoRes, especially on the random benchmark, which tests the algorithms using a variable number of effectors mixing different effector types.

---

> ### Author Response · Authors · 2021-11-24
> **Thank you!**
>
> We would like to sincerely thank Reviewer cvbx for carefully reading our manuscript and response, providing insightful comments and additional clarifications as well as for updating their score.

---

### Decision · Program_Chairs · 2022-01-20

**Decision:**

Accept (Oral)

**Comment:**

This paper proposes a novel representation for pose authoring, and was uniformly lauded by all reviewers.  The AC concurs this paper is far above the threshold for acceptance at ICLR.